# Expertise-driven temporal gaze dynamics during anticipation in volleyball

**Thomas Kanatschnig**[1*], **Živa Korda**[1], **Norbert Schrapf**[2], **Lisa Leitner**[1],
**Christoph Anzengruber**[1], **Otto Lappi**[3], **Christof Körner**[1,4], **Markus Tilp**[2,4],
**Silvia Erika Kober**[1]

**1** Department of Psychology, University of Graz, Graz, Austria, **2** Department of Human Movement Science, Sport and Health, University of Graz, Graz, Austria, **3** Department of Digital Humanities, University of Helsinki, Helsinki, Finland, **4** BioTechMed-Graz, Graz, Austria

\* thomas.kanatschnig@uni-graz.at

## Abstract

Perceptual-cognitive comparisons of experts and novices have consistently shown that experts use specific visual strategies to process visual scenes in their domain of expertise, reflected in eye movement metrics such as fixation rates and durations. We present an investigation of the gaze behavior from professional volleyball players (experts; $n = 14$) during a volleyball anticipation task and compare them to intermediate level volleyball players (amateurs; $n = 25$) and individuals with only basic volleyball experience (novices; $n = 19$). The task consisted of the observation of videos, which were recorded during official national level volleyball matches, each showing unique setting situations. Our results replicate previous findings showing lower fixation rates as well as longer fixation durations in relation to higher expertise. Yet, this trend was only present in the later phase of video observation, i.e., during the course of the rally. In the early phase, i.e., during players' preparation before the service, experts did not differ from amateurs on all fixation metrics, while novices performed comparatively higher rates of fixations. Our findings emphasize the importance of investigating temporal dynamics, as well as using a comprehensive operationalization of perceptual-cognitive processes related to expertise.

## Introduction

Gaze behavior of athletes has long been an intriguing research subject in sports science [1–7]. The investigation of professionals' gaze allows for the identification of search strategies which reflect differences in domain expertise. Team-based sports pose special demands on the scene perception of athletes, when multiple players are on the court at the same time, moving in highly dynamic, but partially predictable patterns. We present the results from our analysis of gaze behavior during an anticipation task in the domain of volleyball, a fast-paced dynamic team sport. In this task,

**Data availability statement:** All relevant data are within the paper and its Supporting information files.

**Funding:** This study was financially supported by the University of Graz for coverage of publication fees. No additional external funding was received for this study.

**Competing interests:** The authors have declared that no competing interests exist.

videos of volleyball game situations were presented, of which the outcome had to be predicted by participants that ranged in expertise from novices (having little relevant domain experience) to professional volleyball players. Additionally, the participants' brain activity and behavioral task performance were measured, for which the results have been reported previously by Kanatschnig et al. [8,9]. In our previous work we showed that volleyball experts and amateurs differed significantly in prediction accuracy, from each other, as well as from novices. This relationship between level of expertise and performance in domain-representative scene perception tasks is a well-established finding in the literature on sports expertise [10–17].

What is more, eye tracking research has consistently shown that experts use specific strategies to actively sample visual scenes in their domain of expertise, reflected in eye movement metrics such as fixation rates and durations [18,19]. A meta-analysis by Mann et al. [19] indicates that experts show lower rates of fixations as well as longer fixation durations compared to non-experts during sport-specific visual tasks, thereby creating the notion of an "expertise-efficiency" relationship. It is hypothesized that experts extract more task-relevant information from a given scene (per fixation and from the visual periphery), which allows them to be more precise in their estimations even with fewer fixations performed. The use of peripheral vision can help to minimize repositioning of the gaze fixation, while still remaining aware of movement happening in the peripheral field [20,21]. Two important concepts in this regard are so-called gaze anchors (cue-optimized fixations in-between relevant sources of information, which allow the simultaneous monitoring of relevant information from peripheral cues), as well as visual pivots (distance-optimized fixations in-between relevant sources of information, aimed at the optimal initiation of saccades to relevant cues) [22]. It has been shown that professional athletes performed substantially better compared to amateurs and non-athletes, during a computerized training intervention, targeted at the optimal use of peripheral vision; in the 3D-MOT task, the ability to optimally distribute the attention between multiple moving targets among distractors – analogous to monitoring multiple players in a volleyball game – is crucial for achieving high performance scores [23]. Furthermore, the concept of knowledge-driven gaze control becomes important, when investigating dynamics of perceptual-cognitive expertise in sports [24]. In this regard, task-related knowledge is arguably one of the most important aspects. Training and repetition generate a "gaze-control strategy" for a specific task, in which search patterns are encoded, aimed at most effectively extracting relevant information from a given visual scene. In other words, experts know what to look for and where to look for it.

Previous research on perceptual-cognitive ability in volleyball has shown that expert players show more advanced visual cue utilization compared to less-skilled players [17,25,26]. Large-scale video footage analyses also revealed patterns which aid the predictability of successful *rallies* (rally: sequence of play from the moment the ball is served until the play ends with a point being scored, i.e., ball hits the ground or goes out of bound), such as the tendency of female teams playing closer to the net [27] or the importance of *reception* and *setting* efficacies (reception: first ball contact of a player following the opposing team's service/attack; setting: second to

last of three allowed ball contacts during a team's ball possession, setting up a player for the attack) [28]. Studies on gaze behavior in volleyball have predominantly incorporated laboratory-based experimental designs and broadly or at least partially supported the notion of an expertise-efficiency relationship [12,17,29,30]. There are also instances of studies where in-situ designs were utilized [31,32]. In a recent study by Zhu et al. [17] the authors found that competitive elite volleyball players used fewer fixations, while also achieving higher performance scores, compared to novices in a spiking anticipation task. Prior to that Piras et al. [29,30] found lower fixation rates as well as longer fixation durations by volleyball experts compared to novices in a volleyball-representative task, where the aim was to anticipate the outcome of setting situations (i.e., predicting the direction of the final pass of the setter player before the attack). The authors attributed these results to expert volleyball players' ability to extract more task-relevant information from each fixation, through greater reliance on visual pivots, enabling the simultaneous extraction of movement information from specific areas of interest (AOI), located predominantly on the body of the observed setter player (i.e., hands, trunk, and legs), allowing for a more efficient initiation of saccades to relevant cues.

Our aim with this work was to investigate expertise-driven temporal dynamics of gaze behavior during anticipation in volleyball. Specifically, we investigated differences in fixation rates and durations during distinct phases of volleyball scene observation. In our experiment, we presented an anticipation task focused on volleyball setting situations, which was performed in a laboratory environment. In the main condition of the task, participants were asked to predict the outcome of prerecorded setting situations from official national level volleyball matches. The inclusion of a control condition (designed to manipulate domain knowledge requirements while keeping perceptual demands identical) provided additional insight into domain-specific effects on gaze behavior. Moreover, our sample consisted of three distinct groups in terms of volleyball expertise. We incorporated an intermediate level amateur group in addition to novices and experts, allowing for a more fine-grained assessment of expertise-related aspects of gaze behavior. In combining these aspects, we see the greatest potential of our investigation for providing a comprehensive picture of dynamic perceptual-cognitive processes underlying anticipation in volleyball. As this work was planned as an explorative investigation (see https://osf.io/exhr3) no specific a-priori hypotheses were defined. Yet, we expected to find lower fixation rates as well as longer fixation durations in relation to higher expertise, as it was found in previous investigations [12,29,30].

## Method

As Mann et al. [19] stated, several factors of the experimental research design can significantly moderate the relationship between sport expertise and gaze behavior. Key aspects that we emphasized in our investigation were the following: (i) To evaluate volleyball expertise on a spectrum, three groups of participants with different levels of volleyball expertise were recruited, i.e., *novices, amateurs* and *experts*. (ii) The video material we used were recordings from official national level volleyball matches, depicting *naturalistic volleyball setting situations*. (iii) We examined *two time intervals of interest*, i.e., *before* and *during* the rally. (iv) In addition to the main task condition, aimed at the *prediction* of setting situations, we also presented a *control* condition to differentiate between domain-specific and domain-general aspects of gaze behavior. In the following, we explain how these aspects were implemented into our experimental design. Furthermore, we provide explanations below for all relevant items in the reporting guideline checklist for eye-tracking research by Dunn et al. [33] (see sections "Apparatus and procedure" and "Preprocessing").

### Participants

In total 64 participants took part in the study. Six participants had to be excluded from the analysis due to eye-tracking calibration issues, leaving the data of 58 participants. Among them were 34 women and 24 men, ranging in age from 18 to 42 years ($M = 23.19$ years, $SD = 4.61$). All participants reported to have normal or corrected to normal vision. Contact lenses were worn if vision aid was necessary (one person wore hard lenses, 15 wore soft lenses).

To categorize participants based on their volleyball expertise we defined operational criteria to allocate participants to one of three expertise groups. From the lowest to highest level of expertise, the groups were named "Novice", "Amateur" and "Expert" group, respectively. The criteria for the allocation of participants were as follows:

- The Novice group ($n = 19$; 13 women; $M = 23.63$ years, $SD = 3.22$) included individuals who reported only occasional volleyball experience.

- The Amateur group ($n = 25$; 12 women; $M = 22.72$ years, $SD = 4.60$) consisted of individuals who did not play professionally but were actively playing in a volleyball club or in a university volleyball course at intermediate or advanced level.

- The Expert group ($n = 14$; 9 women; $M = 23.43$ years, $SD = 6.25$) consisted of professional volleyball players (and one professional coach) who actively played (or coached) in the first or second national volleyball league of Austria.

To measure prior volleyball experience, participants were asked to report on the approximate *lifetime number of years* and the average number of *weekly hours* of volleyball playing. Participants were also asked to report information about other sport-specific habits. Comparative statistics for the main volleyball- and sports-specific as well as socio-demographic variables were calculated by means of $\chi^2$-tests of independence plus Cramer's $V$ in the case of categorical variables, and between-subjects analyses of variance (ANOVA) plus partial $\eta^2$ in the case of continuous variables (see Table 1). All participants gave informed written consent before participation. Recruitment and data collection was performed from September 1, 2022, until February 28, 2023, via contact with local volleyball clubs, as well as through advertisement on social media platforms, e-mail distribution and flyer postings. All procedures have been conducted in accordance with the Declaration of Helsinki and were approved by the ethics committee of the University of Graz (GZ. 39/105/63 ex 2021/22).

**Table 1. Comparative statistics for main participant variables.**

| Variable: | Novice (n=19): | Amateur (n=25): | Expert (n=14): | Statistics: | | | | |
|---|---|---|---|---|---|---|---|---|
| *Categorical* | | | | df | $\chi^2$ | p | V | Sig. |
| Sex (female/male) | 13/6 | 12/13 | 9/5 | 2 | 2.10 | .350 | .19 | |
| "Do you currently play actively in a volleyball club?" (no/yes) | 19/0 | 15/10 | 0/14 | 2 | 33.3 | <.001 | .76 | *** |
| "Do you play volleyball in your free time?" (no/yes) | 9/10 | 0/25 | 0/14 | 2 | 21.9 | <.001 | .61 | *** |
| "Are you currently attending one or more volleyball courses at the University Sports Institute (USI)?" (no/yes) | 19/0 | 9/16 | 14/0 | 2 | 29.2 | <.001 | .71 | *** |
| "Apart from volleyball, do you play sports regularly?" (no/yes) | 4/15 | 7/18 | 0/14 | 2 | 4.66 | .097 | .28 | |
| *Continuous* | | | | df | F | p | $\eta p^2$ | Sig. |
| Age in years | 23.63 ±3.22 | 22.72 ±4.60 | 23.43 ±6.25 | 2, 55 | 0.23 | .796 | .008 | |
| "On average, how many hours a week do you play volleyball? (Please provide an approximate estimate of the number of hours.)" | 0.55[ab] ±0.83 | 3.66[ac] ±2.03 | 9.57[bc] ±3.01 | 2, 55 | 79.71 | <.001 | .743 | *** |
| "Approximately how many years of volleyball experience do you have? (Please provide a rough estimate of the years.)" | 3.03[ab] ±3.75 | 5.52[ac] ±3.31 | 11.29[bc] ±3.75 | 2, 55 | 22.23 | <.001 | .447 | *** |
| "On average, how many hours per week do you exercise? (Please provide an approximate estimate of the number of hours.)" | 4.32[a] ±2.22 | 5.36 ±2.46 | 9.18[a] ±6.18 | 2, 55 | 7.72 | .001 | .219 | ** |

Absolute participant counts are presented for categorical variables. Means and standard deviations are presented for continuous variables. Variables with significant overall test results are indicated with asterisks (**: $p < .01$; ***: $p < .001$). For continuous variables, significant pairwise comparisons, as determined by Benjamini-Hochberg corrected $t$-tests, are indicated with uppercase letters ([a-c]).

## Stimuli

The task we presented to participants was a scene perception/anticipation task, in which participants were asked to watch videos of volleyball setting situations and give a prediction about the outcome of these situations. The videos included in this experiment have been first used by Schrapf et al. [34] in earlier research on volleyball coaches' anticipation ability.

The task consisted of 63 unique video stimuli, all showing a common game situation in 6 vs. 6 women's indoor volleyball, in which the *service player* (service player: player who starts a rally by playing the ball to the opposing team, referred to as the service/serve) of one team prepared for and subsequently played the service. After the opposing team received the ball, the *setter player* (setter player: player who plays the final pass, the setting/set, to a teammate for the attack) of the team played the pass for the ensuing attack. For the experiment, the videos were prepared so that they always stopped 0.08 s (two frames) before the setter touched the ball, i.e., the setter's pass destination was occluded.

The videos varied in length between 3 and 13 s and were recorded with a frame rate of 25/s and a resolution of 720x576 px (5:4 aspect ratio) during official games of the 2nd Austrian Women's Volleyball League. Each video consisted of two main phases of interest. The "Preparation" phase lasted 0–10 s ($M = 5.24$ s, $SD = 2.66$) from the start of the video until the service player served the ball. The variation in length was due to the differing preparation times the service players took before serving the ball in each video. While this range of different preparation lengths potentially influences gaze behavior, it resembles real match situations, where players have the freedom to prepare the start of the rally for a certain amount of time. The Preparation phase was followed immediately by the "Rally" phase, which was set to 3 s and included the course of the rally from the service until the end of the video. Given the similar structure of each rally sequence, the 3 s time interval very accurately covered the rally in each respective video. The rationale behind the separation into Preparation and Rally phases was based on the idea that gaze patterns before and during the rally may vary, e.g., because of the movements of players on the court, which are rather stationary before the service and become active after the service.

Furthermore, there were two different task conditions. In the "Prediction" condition, participants were asked to indicate the location to which they believe the setter would pass the ball ("Where will the SETTER pass the ball to?"). In Prediction trials participants were presented with four options of possible destinations of the setter's pass. In the "Control" condition, the task was simply to name the location of the service player at the time of service ("Where does the SERVICE player stand when serving?"). In Control trials participants had the option between three possible service locations. For an illustration of the response screens, which depicted the respective response options per condition, see Fig 1. In the Prediction condition participants had to actively observe the full course of the rally to make a well-informed judgement about its outcome (i.e., setter's pass), whereas in the Control condition there was no need to actively observe the rally, as the crucial information (i.e., service location) was presented right at the beginning of each video. Therefore, domain-specific aspects, i.e., expertise-driven deliberation processes for the accurate prediction of the setting, should theoretically be more prevalent in the Prediction condition compared to the Control condition, where mainly domain-general aspects, i.e., general perception of players actions with the simple goal of memorizing a player's position on the court, should determine participants' gaze behavior.

## Apparatus and procedure

For the recording of eye movements, participants were asked to position their head on a chin rest at a fixed distance of 620 mm from a 24.4-inch 60 Hz LCD screen with a resolution of 1920x1080 px (16:9 aspect ratio). Monocular eye-tracking (infrared, video-based) of the right eye was performed at 500 Hz using an EyeLink 1000 system (SR Research Ltd.) in desktop mode, placed under the presentation screen. Gaze recording (x/y-coordinates in pixel) was performed using a dedicated host PC running SR Research's control and recording software (v4.56). Testing took place in a sound attenuated and electrically shielded laboratory cabin. Lighting in the cabin was reduced to a dimmed white light emitted by LED strip lights installed behind the participant, facing the rearward cabin wall, resulting in a subtle illumination of the cabin.

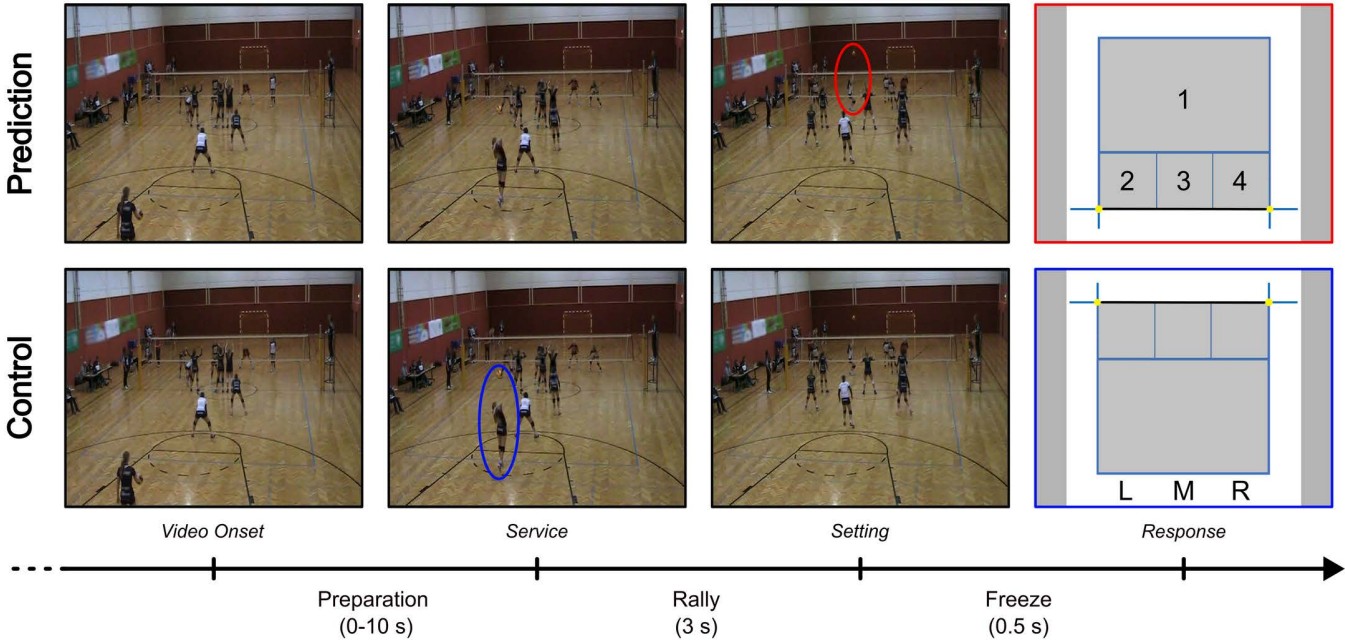

**Prediction:** *"Where will the SETTER pass the ball to?"* (position 1, 2, 3 or 4)
**Control:** *"Where does the SERVICE player stand when serving?"* (left [L], middle [M] or right [R])

**Fig 1. Visualization of the volleyball anticipation task.** The task included Prediction and Control conditions, the order of which was counterbalanced across participants. The two main phases of interest for the analysis of gaze behavior were the Preparation phase and Rally phase. The time from video onset until the service marked the Preparation phase (0-10 s), while the time from the service until the end of the video marked the Rally phase (3 s). The Freeze phase (0.5 s) marked the visibility of the last video frame before the appearance of the response screen, which prompted participants to give their response to the respective task condition. The main players of interest, namely the setter in the Prediction condition and the service player in the Control condition, are highlighted in red and blue, respectively. Image material partially reprinted from Kanatschnig et al. [8] under a CC BY license, original copyright 2025.

All 63 video stimuli were shown in randomized order once during Prediction and once during Control, meaning the visual input was the same during both task conditions and resulting in a total of 126 trials. Prediction and Control trials were presented together per condition. Additionally, task trial presentation was divided into two separate blocks. In the first block, participants received trials where the actions of interest (i.e., the setting in the Prediction condition and the service in the Control condition) happened on the nearer side of the court (court side in front of the net), whereas in the second block the actions occurred on the farther side of the court (court side behind the net), with respect to the camera's field of view of the video recordings. Within presentation blocks, the order of Prediction and Control conditions was counterbalanced across participants but was the same in both blocks for each individual participant, i.e., if a participant started the first block with Prediction (or Control), they also started the second block with Prediction (or Control). Standard 9-point calibration and validation (as implemented in the eye-tracking recording software) were performed at the beginning and the middle of each presentation block. Drift correction was performed before each trial. At the start of each block participants carried out practice runs (using different video stimuli), to familiarize themselves with the respective task condition requirements.

The presentation of all trials from the volleyball anticipation task followed the same structure. Before each trial, participants centered their gaze on a fixation disk in the middle of the screen for drift correction, which was followed by another 2 s of fixation. Next, the trial started with the Preparation phase followed by the Rally phase of the respective video. After the Rally phase finished, the last video frame remained visible for 0.5 s (i.e., the "Freeze" phase; a deliberately chosen

design element for the investigation of brain activity, see [8]) and was followed by the appearance of the response screen depicting the schematic outlines of a volleyball court with the respective answer options, which depended on the task condition at hand. After participants gave their response via button press on a computer keyboard, a randomized inter-trial-interval of 3–4 s occurred showing a blank grey screen. The implementation of the task was carried out with PsychoPy (v2021.2.3, [35]). A visual representation of the procedure used during the volleyball anticipation task is presented in Fig 1.

## Preprocessing

Gaze data was retrieved through the EyeLink Data Viewer software (v4.3.210, SR Research Ltd.) and further processed using R (v4.4.3, [36]), RStudio (v2024.12.1 + 563, [37]) and the *tidyverse* package (v2.0.0, [38]). For data classification we followed the criteria used by Walcher et al. [39]. Blinks detected by the eye-tracker (i.e., successive samples where no gaze position was detected) were extended by 100 ms before and after the blink period to account for partial lid closure. Saccades were classified as successive samples with velocity values larger than 30°/s or acceleration values larger than $8000°/s^2$ and if the event lasted at least 6 ms. Microsaccades, classified as saccades with an amplitude equal or lower than 1° visual angle, were excluded from the analysis. All sample sequences which were not blinks or saccades were classified as fixations. Therefore, fixation durations were defined as the timeframe from the onset of a fixation until the time point when a new saccade or blink occurred, which means that the last fixation of a phase stretched into the subsequent phase. To maintain temporal alignment with the cognitive trigger causing a fixation, fixations that occurred during phase transitions were assigned to the phase of their onset.

The saccade classification criteria we used were derived from the eye-tracker manufacturer's (SR Research Ltd.) implemented criteria for automatic saccade identification, which have shown to be well suited for tasks in lab-based settings [39–41]. However, because our stimuli included moving targets, e.g., the ball, target motion speed had to be considered in our classification. With our analysis, we did not investigate smooth pursuit movements, which made it necessary to define how eye movements related to the tracking of moving targets were classified. We approached this by identifying the fastest ball movements in the videos, which coincided with the service at the beginning of each video. As the service is generally played very strongly in volleyball, the ball moves at a high speed from one side of the court to the other. With a subsample of eight videos, in which we visually identified the fastest serves across all videos, we calculated the Euclidean distance and time the ball traveled (on screen) from when it left the hand of the service player until its reception on the other side of the court. The average ball speed during service was 148 mm/s, while 199 mm/s was the fastest ball speed across the eight videos. In terms of visual angle (at a 620 mm distance from the screen) the average ball speed equaled 14°/s, while the fastest speed equaled 18°/s, meaning that the fastest movement presented on screen was still under the saccade velocity threshold of 30°/s. Therefore, we can say that movements related to target tracking on screen, such as following the ball, were generally classified as sequences of fixations and small intermediate saccades.

The fixation and saccade data were cleaned before the statistical analysis according to the following criteria. The data was first filtered by time stamps to only include fixations and saccades that happened during the Preparation and Rally phases of the volleyball anticipation task. Fixations with an estimated gaze point outside the screen dimensions as well as ones that duration-wise exceeded the end of a trial were excluded (0.23%). Saccades that reached an amplitude that was equal to or higher than the diagonal of the presentation screen (53°) as well as ones that lasted longer than 200 ms were excluded (0.07%). Remaining trials where more than 50% of gaze data was missing (due to blinks or non-detected gaze) were excluded, which occurred in the data of seven participants. Only 23 trials from these seven participants had to be removed from the data in total (0.31%).

## Statistical analysis

Our main dependent variables (DV) *fixation rate* (number of fixations per second) and *fixation duration* (mean fixation duration in milliseconds) were analyzed using linear mixed effect models (LMM) by means of the *lme4* (v1.1-36, [42]) and

*lmerTest* (v3.1-3, [43]) packages for R. Log-transformed values were used for the LMM analysis of fixation duration, as we identified the issue of strong skewness of the fixation duration data. For the sake of interpretability untransformed fixation duration millisecond values are presented for descriptive statistics and visualization in the "Results" section. Analogous to fixation rate we additionally investigated saccade rate. However, due to the high correspondence between the two metrics, the saccade rate results did not provide relevant additional information and are therefore presented under "Supporting information" (see Tables B & C in S6 Table). The same LMM structure was used for each DV, which was defined by the following formula:

$$DV \sim Group \times Task \times Phase + (1 \mid Code) + (Task \times Phase \mid Video) \tag{1}$$

In this formula *Group* (i.e., Novice vs. Amateur vs. Expert group), *Task* (i.e., Prediction vs. Control task condition), and *Phase* (i.e., Preparation vs. Rally phase) signify the included fixed factors. As for random factors (1 | *Code*) signifies the inclusion of random intercepts for each participant, while (*Task × Phase | Video*) signifies the inclusion of random intercept as well as interaction slopes of *Task* and *Phase* for each stimulus *Video*. The rationale behind this random effect structure was to account for general baseline differences between participants through the inclusion of the random intercept term for participant *Code*, and more specifically the influence that visual stimulus properties exerted on DVs through the inclusion of the random intercept and interaction slopes term of *Task* and *Phase* for stimulus *Video*. For instance, we observed that fixation rates were systematically influenced depending on how near or far actions of interest within different levels of *Task* and *Phase* occurred across videos. The nearer/farther to the camera the observed actions happened, the higher/lower were generally fixation rates, presumably due to the generally wider/shorter visual angles of observed players, objects and distances between them, making it necessary to switch fixation positions more/less frequently. Therefore, by allowing the effect of *Task* and *Phase* as well as their interaction to vary across videos we specifically account for the occurrence of such random effects within our model. This is important, as without specifically accounting for these effects, the statistical results could be distorted.

Global fixed effects were tested with a Type III ANOVA. To identify differences between novices, amateurs and experts we calculated pairwise comparisons for each combination of *Task* and *Phase* for each DV using the *emmeans* (v1.10.7, [44]) package. The *rstatix* (v0.7.2., [45]), *car* (v3.1-3, [46]) and *e1071* (v1.7-16, [47]) packages were additionally used for certain statistical calculations during our analyses. The *ggpubr* (v0.6.0, [48]) and *sjPlot* (v2.8.17, [49]) packages were used for visualizations. LMM degrees of freedom were calculated using Satterthwaite's method [50]. Model diagnostics were calculated for each DV, which did yield some deviations regarding the normality assumption, yet to no extreme extent. To control the false discovery rate, Benjamini-Hochberg-adjusted *p*-values were calculated for pairwise comparisons, thereby reducing the Type I error probability [51]. The effect sizes provided for pairwise comparisons, which were computed using the *eff_size* function of *emmeans*, can be interpreted as an estimation of Cohen's *d*. The significance level for all analyses was set to $\alpha = 0.05$ (two-tailed). All main and complementary analyses including model diagnostics can be examined using the provided data files and R code under "Supporting information".

## Results

As gaze strategies were expected to differ depending on task condition and video phase, we looked at gaze distributions, i.e., fixation density heatmaps, to see what the participants' main areas of focus during the completion of the task were. Fig 2 shows an example of the gaze distributions during the completion of the Prediction and Control conditions for one selected video of the volleyball anticipation task in relation to phase. The density patterns presented in Fig 2 are an accumulation of all fixations (no saccades included) across all participants, as we identified through visual inspection that the general gaze distributions did not differ greatly between groups. Descriptively, during the Preparation phase, there was a noticeable difference between the two task conditions, depicted by a stronger aggregation of fixations on the players near

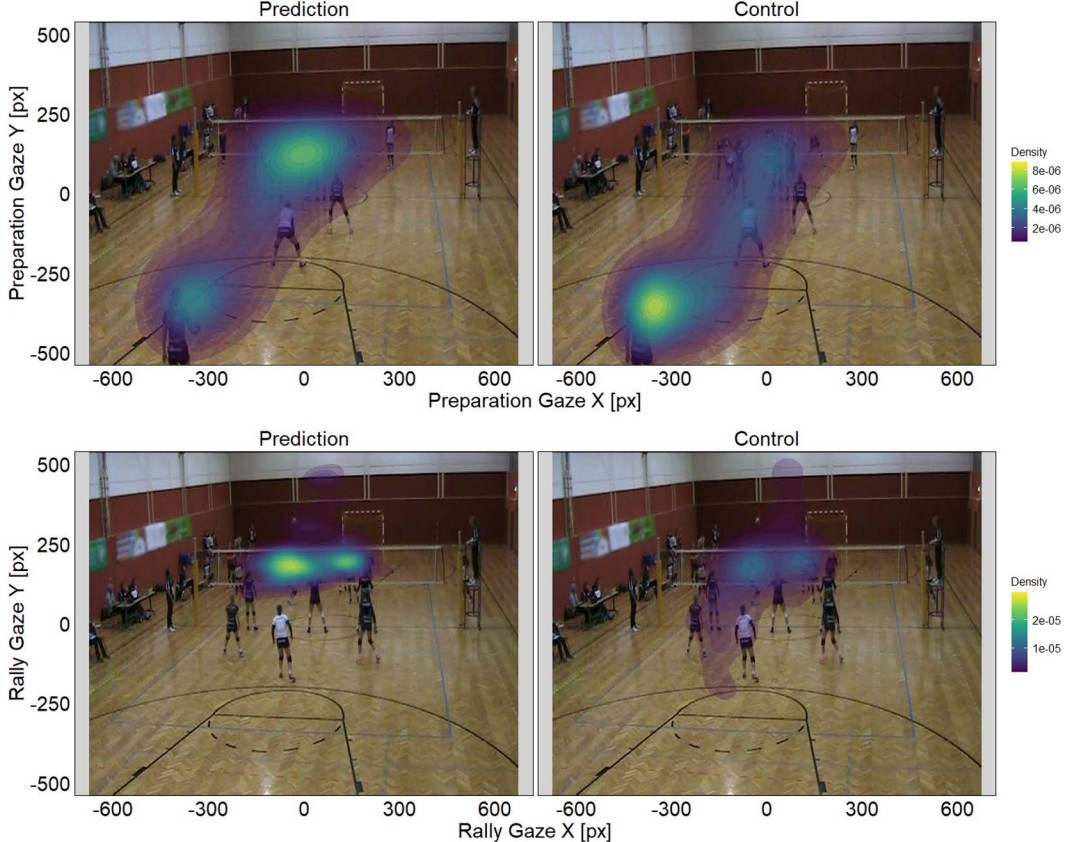

**Fig 2. Gaze distributions per phase (example).** Fixation density heatmaps from all participants for each task condition, i.e., Prediction and Control, during the completion of the Preparation phase (upper panels) and Rally phase (lower panels) of one example video stimulus. For the purpose of illustration, the background image in the Preparation panels shows the first frame, while the image in the Rally panels shows the last frame of the example video. Color patterns represent estimated probability density of gaze points falling in a respective region, with warmer colors indicating areas where participants looked more frequently. Note that density patterns resulted from the aggregation of all fixations performed during the entirety of the respective phase and are comparable across conditions within that phase. Different density ranges are a result of the different total amounts of fixations performed in each phase. Image material partially reprinted from Kanatschnig et al. [8] under a CC BY license, original copyright 2025.

the net during Prediction, while during Control the focus lay more on the service player. During the Rally phase, there was a shift of attention towards the players on the other side of the net in both conditions, but there was a higher density of fixations observable during Prediction, compared to Control. As this example demonstrates, there are noticeable global differences of the participants' attentional focus, in relation to task condition and phase.

Full ANOVA results for global fixed effects from the LMM analyses of the main DVs, i.e., fixation rate and fixation duration, are presented in Table 2. In addition to other significant main and interaction effects, the three-way interaction *Group × Task × Phase* yielded a significant result in the case of fixation duration ($p = .010$), while the *p*-value for fixation rate is on the verge of significance ($p = .063$). Given these results, we performed subsequent pairwise group comparisons to identify differences between expertise groups for each factor level combination of *Task* and *Phase*.

Pairwise comparisons revealed that the Novice group ($M = 1.65$, $SD = 0.34$) made significantly more fixations than the Amateur group ($M = 1.45$, $SD = 0.34$; *t-ratio*(62.40) = 2.65, $p = .031$, $d = 0.24$) in the Prediction condition during Preparation. Furthermore, the Novice group ($M = 1.46$, $SD = 0.24$) made significantly more fixations than the Expert group ($M = 1.21$, $SD = 0.18$; *t-ratio*(62.42) = 2.95, $p = .013$, $d = 0.31$) in the Prediction condition during Rally. Fig 3 provides a visualization of group differences of fixation rate.

**Table 2. ANOVA results for LMM analyses of main fixation metrics.**

| | Fixed Effect: | SumSq | MeanSq | df | F | p | Sig. |
|---|---|---|---|---|---|---|---|
| **Fixation Rate:** | Group | 1.63 | 0.81 | 2, 55.01 | 2.79 | .070 | |
| | Task | 0.15 | 0.15 | 1, 62.26 | 0.51 | .480 | |
| | Phase | 1.66 | 1.66 | 1, 60.67 | 5.68 | .020 | * |
| | Group × Task | 2.78 | 1.39 | 2, 13921.12 | 4.77 | .008 | ** |
| | Group × Phase | 11.36 | 5.68 | 2, 13921.06 | 19.49 | <.001 | *** |
| | Task × Phase | 6.01 | 6.01 | 1, 54.87 | 20.61 | <.001 | *** |
| | Group × Task × Phase | 1.61 | 0.81 | 2, 13921.11 | 2.76 | .063 | |
| **Fixation Duration:** | Group | 1.02 | 0.51 | 2, 55.01 | 2.15 | .127 | |
| | Task | 0.15 | 0.15 | 1, 63.44 | 0.64 | .426 | |
| | Phase | 0.47 | 0.47 | 1, 61.13 | 1.97 | .165 | |
| | Group × Task | 4.49 | 2.25 | 2, 13920.71 | 9.42 | <.001 | *** |
| | Group × Phase | 13.61 | 6.80 | 2, 13920.45 | 28.52 | <.001 | *** |
| | Task × Phase | 28.45 | 28.45 | 1, 65.44 | 119.26 | <.001 | *** |
| | Group × Task × Phase | 2.19 | 1.09 | 2, 13920.61 | 4.58 | .010 | * |

Type III ANOVA results with degrees of freedom calculations using Satterthwaite's method. Significant fixed effects are indicated with asterisks (*: $p < .05$; **: $p < .01$; ***: $p < .001$).

Furthermore, pairwise comparisons revealed that the Expert group showed significantly longer fixations ($M = 1042$, $SD = 227$) than the Amateur group ($M = 876$, $SD = 180$; t-ratio(62.43) = −2.25, $p = .042$, $d = −0.27$) and the Novice group ($M = 827$, $SD = 175$; t-ratio(62.41) = −2.90, $p = .015$, $d = −0.36$) in the Prediction condition during Rally. Note that statistical calculations were performed on log-transformed fixation duration values, while descriptive statistics are presented as untransformed millisecond values for better interpretability. Fig 4 provides a visualization of group differences of fixation durations. For full descriptive statistics see Table 3 (for full pairwise comparison results see Table A in S6 Table).

## Discussion

We investigated the gaze behavior of three groups with distinct levels of volleyball expertise, i.e., novices, amateurs, and experts, during the performance of a volleyball-specific anticipation task. Videos of setting situations from professional 6 vs. 6 women's volleyball matches were presented to participants, while their eye movements were recorded via eye-tracking. The participants' aim in the main Prediction condition was to predict the outcome of each setting situation, while the Control condition served as a reference for the investigation of task-related effects. Prior research [18,19] showed that fixation rate and duration are strong indicators of experts' efficiency of gaze behavior in sports-related performance measures. Thus, our analysis focused on fixation rate and duration. For the analysis, we defined two specific phases of interest, i.e., the Preparation phase (time from video onset until the service), as well as the Rally phase (time from the service until the end of the video, including the rally itself). With our findings we present further evidence of the inherent relationship between sports expertise and gaze efficiency, as well as the role of temporal dynamics of experts' perceptual-cognitive processing.

An interesting pattern emerged from the results of the Preparation phase. We found that novices made significantly more fixations compared to amateurs in the Prediction condition. No significant group differences were found for fixation duration. Interestingly however, the experts' average descriptively lay between that of novices and amateurs on all dependent measures in Prediction during Preparation, albeit more closely to that of amateurs (see Figs 3 and 4). This implies that experts' fixation rates were more comparable to that of amateurs, but still fewer fixations were performed compared to novices. The similarity to the lower-level Amateur group suggests that the Expert group performed a more exploratory

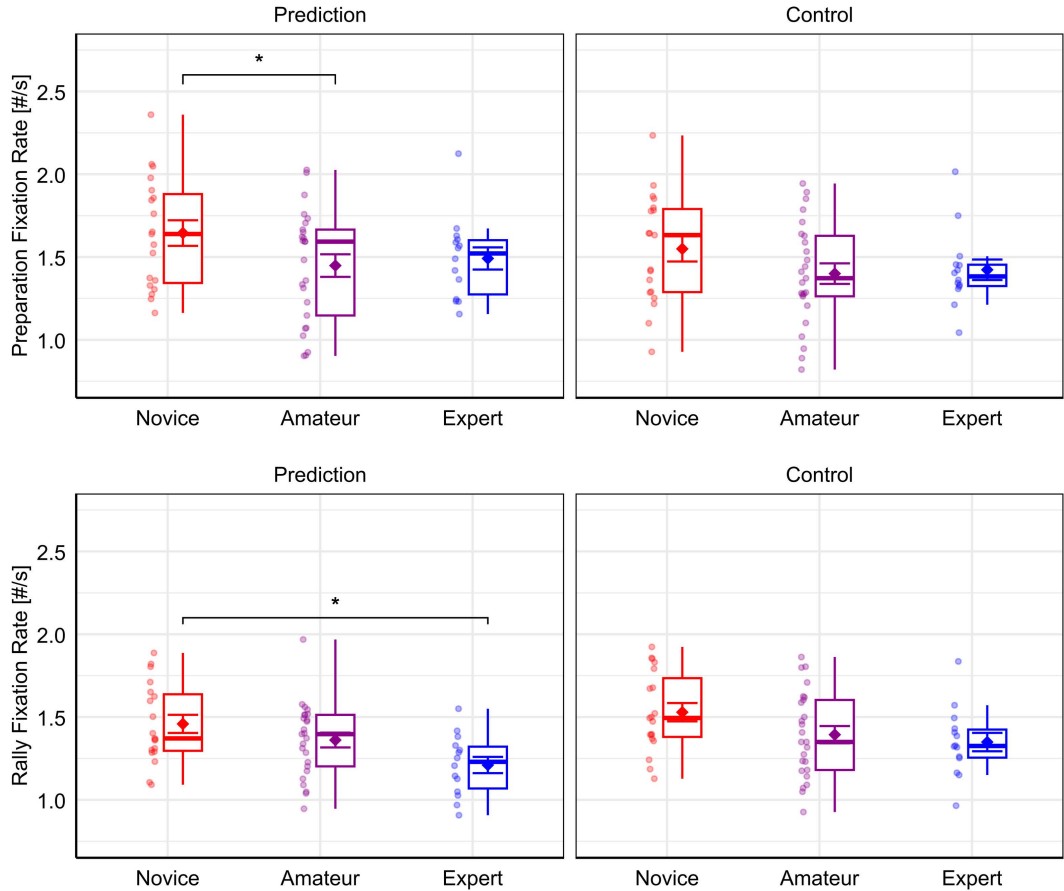

**Fig 3. Results for the analysis of fixation rate.** Results for fixation rate for each task condition, i.e., Prediction and Control, as well as expertise group, i.e., Novice, Amateur, and Expert, during the Preparation phase (upper panels) and Rally phase (lower panels), respectively. Note that jitter dots beside boxplots represent individual participants of the respective groups. Diamond symbols with error bars represent the mean and standard error. Significant pairwise group differences, as determined through LMM analysis, are indicated with asterisks (*: $p < .05$).

gaze strategy during the Preparation phase. This was done possibly to extract more task-relevant information helping them to form an idea of the situation's outcome already at an earlier stage of video observation. This interpretation is in line with the notion that experts make use of advanced perceptual cues helping them to make quicker decisions about their opponent's actions [24–26]. Analogous to squash experts' ability to extract relevant information for the prediction of an opponent's stroke earlier in the movement cycle [10], volleyball experts appear to perform more fixations before the start of the rally to extract more information for the prediction of the ensuing setting.

Transitioning to the Rally phase, we see a different pattern emerging from the data, which is more in line with the overall findings of previous research [18,19]. This is illustrated by what can be described as a linear trend of higher expertise relating to more efficient gaze behavior, i.e., relatively lower fixation rates as well as longer fixation durations from experts. While visually processing the course of the rally, experts showed a significantly lower fixation rate compared to novices in the Prediction condition. Given the inherent relationship between fixation rate and duration, the analysis of fixation duration further corroborates this pattern, depicting the same linear trend in the reversed direction. Experts showed significantly longer fixation durations compared to novices and in this case also amateurs, in the Prediction condition during Rally.

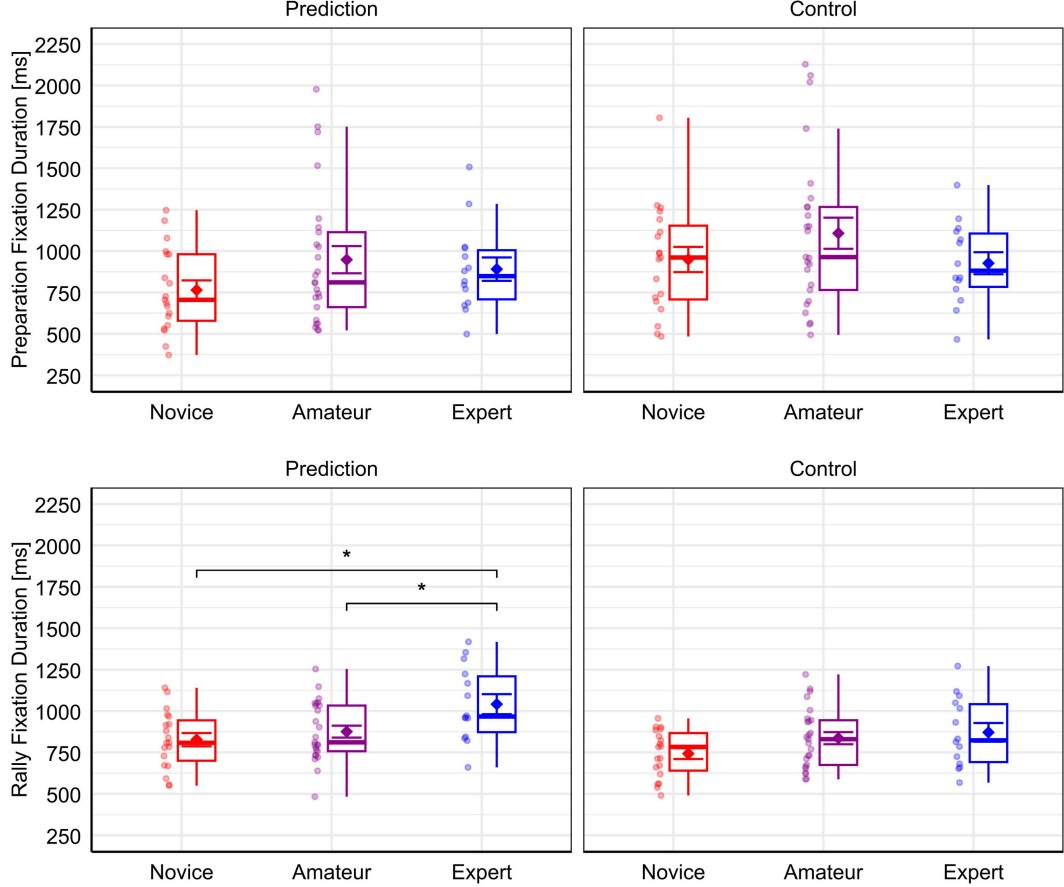

**Fig 4. Results for the analysis of fixation duration.** Results for fixation duration for each task condition, i.e., Prediction and Control, as well as expertise group, i.e., Novice, Amateur, and Expert, during the Preparation phase (upper panels) and Rally phase (lower panels), respectively. Note that jitter dots beside boxplots represent individual participants of the respective groups. Diamond symbols with error bars represent the mean and standard error. Significant pairwise group differences, as determined through LMM analysis on log-transformed values, are indicated with asterisks (*: $p < .05$).

**Table 3. Descriptive statistics for main fixation metrics.**

| | Phase: | Task: | Novice ($n = 19$): | | Amateur ($n = 25$): | | Expert ($n = 14$) | |
| --- | --- | --- | --- | --- | --- | --- | --- | --- |
| | | | *M* | *SD* | *M* | *SD* | *M* | *SD* |
| **Fixation Rate:** | Preparation | Prediction | 1.65 | 0.34 | 1.45 | 0.34 | 1.49 | 0.25 |
| | | Control | 1.55 | 0.33 | 1.40 | 0.31 | 1.42 | 0.23 |
| | Rally | Prediction | 1.46 | 0.24 | 1.36 | 0.22 | 1.21 | 0.18 |
| | | Control | 1.53 | 0.24 | 1.39 | 0.26 | 1.35 | 0.21 |
| **Fixation Duration:** | Preparation | Prediction | 765 | 253 | 948 | 410 | 891 | 264 |
| | | Control | 949 | 331 | 1107 | 470 | 926 | 249 |
| | Rally | Prediction | 827 | 175 | 876 | 180 | 1042 | 227 |
| | | Control | 743 | 142 | 836 | 182 | 871 | 212 |

Means and standard deviations for main fixation metrics are presented for each group within the respective group columns (Fixation Rate: number of fixations per second; Fixation Duration: mean fixation duration in milliseconds).

In general, our results are in line with previous investigations of gaze behavior in the domain of volleyball such as the works by Piras et al. [29,30] who found that expert volleyball players showed lower fixation rates as well as longer fixation durations compared to novices, during a setting anticipation task. There are strong similarities in the experimental design to our investigation which makes these previous findings very comparable to ours, albeit first and foremost to the results of the Rally phase. As Piras et al. [29,30] analyzed gaze behavior during the observation of dynamic setting sequences, this is more comparable to the Rally phase of our investigation. Notably, a new finding of our investigation is that the inclusion of a third group of intermediate expertise level, i.e., the Amateur group, further corroborates the relationship of fixation rate/duration and expertise. This suggests that the extent of prior experience serves as a reliable predictor of perceptual-cognitive efficiency.

Furthermore, the inclusion of the Control condition served as an additional source of information concerning task-related effects. We did not find significant group differences for fixation rate and duration in the Control condition. The absence of group differences for fixation metrics in Control compared to Prediction, signify the existence of general differences in gaze behavior in relation to task condition demands. We assume that the higher prevalence of domain-specific demands (i.e., deliberation for setting prediction) in the Prediction condition has led to higher precision in the differentiation between the three expertise groups, compared to the Control condition, which consisted mainly of domain-general demands (i.e., general perception of players' actions). On a behavioral performance level, this is in accordance with Kanatschnig et al. [8], where we reported that higher expertise was related to higher task performance in Prediction but not in Control; overall we found that experts achieved 67.0% task accuracy in Prediction, while amateurs and novices only achieved 45.1% and 35.8% accuracy, respectively. All three groups differed significantly from each other on task accuracy in Prediction, while no significant group differences were observed in Control (expert: 90.0%, amateurs: 90.3%, novices: 84.4%).

According to Mann et al. [19], the strategy of performing fewer fixations signifies more efficient information extraction per fixation by experts compared to non-experts. However, our findings suggest that such a linear relationship between higher expertise and more efficient gaze behavior is not necessarily given during the entirety of the information extraction process. While we found support for an expertise-efficiency relationship when participants were observing the Rally phase, which is in line with similar previous investigations [29,30], we found a different pattern when the participants observed the Preparation phase before the start of the rally. During the Preparation phase, the experts showed more exploratory gaze behavior, signified by relatively higher fixation rates, making them more comparable to the amateurs at that stage. Our interpretation is that experts were able to acquire more task-relevant information before the start of the rally, guided by their greater domain-specific task knowledge [24]. This advance in knowledge by experts could have led to more efficient gaze coordination during the more dynamic Rally phase, as task-relevant information has already been encoded before.

In addition to faster visual cue utilization [10,17,24–26] in the early stage, the use of peripheral vision could also have contributed to experts' efficiency in information processing [20,21]. The advantage of having essential task-relevant information already encoded, such as the positioning of key players on the court, would make it irrelevant to fixate on these locations again, as long es no movement occurs. As soon as the Rally phase started, multiple players began moving at once, making the use of visual pivots [22,29,30] located centrally in the zone of action a viable technique to staying aware of movement of multiple players, thereby minimizing fixation rates in this phase. We assume that experts may have used such visual pivots, which could have contributed to their greater efficiency during the Rally phase. Moreover, their experience might have led experts to intuitively observe the area near the net more closely, as it has been shown that female volleyball players are generally playing closer to the net during settings [27], thereby further narrowing down the area of interest.

## Limitations

It is necessary to emphasize that we were not only able to observe these results through the distinction between Preparation and Rally phases, but also through the inclusion of amateurs as an intermediate level group in our experimental

design. Furthermore, through the inclusion of a control condition, we were able to give a broader insight into expertise-dependent temporal dynamics of anticipation processes in the domain of volleyball. Nevertheless, our study did not come without certain limitations.

As the present work was planned as an explorative investigation, no dedicated a priori power analysis was performed for the analysis of gaze metrics. However, we found that sample sizes of previous studies (primarily with respect to numbers of professional/expert individuals) were mostly comparable to ours [14,25,29,30]. Also, although there is an imbalance in our group sizes, we do not assume that this circumstance gravely affected our results, as LMMs were found to be quite robust against sample imbalances [52] and because we gathered multiple data points from each participant, which has been shown to improve statistical power [53]. With regard to our stimulus material, by presenting scenes of setting situations from official national level volleyball matches we deem our stimulus material ecologically valid, in the sense that it reproduces temporal dynamics of setting situations how they appear in real matches. However, given our laboratory setup, the stimulus material was presented on a computer screen and our participants were not able to move their head freely, which certainly were no naturalistic conditions. It is possible that gaze behavior and the use(fulness) of different types of peripheral visual information were affected by these viewing conditions. Investigations in naturalistic settings have a clear advantage in that regard [3,31,32]. Also, the use of monocular eye-tracking in our setup (chosen primarily to enable a simpler calibration procedure better suited for our co-registration design with EEG, see [8]) lacks the possibility to investigate more detailed information regarding depth perception and peripheral vision. Our analysis also lacks a differentiation between experts' primary playing positions, which was shown to influence visual search behavior [12]. Further, we did not investigate gaze metrics with respect to specific AOI, as done in previous studies [29,30]. In many sequences, players were partially or fully occluded by other players, therefore gaze fixations could not be uniquely attributed to specific players or interest areas. While in principle spatial analyses provide a more detailed picture regarding visual search strategies, in this case we decided to omit such analyses to avoid introducing misleading conclusions. Also, given the circumstance that the video material was recorded during women's volleyball matches from an off-court perspective, participants had to adapt to these viewing conditions, which were less natural than an on-court/ego perspective [17], and there is the potential that male participants differed from female participants in their gaze behavior, as male volleyball players are expected to be not as familiar with women's volleyball sequences, yet the general play sequences are still very similar to men's matches. Lastly, as during the experiment each video was shown twice to participants, i.e., once during Prediction and once during Control, potential learning effects must be considered with regards to familiarity with the video content. However, we do not assume that this impacted our overall results, as we utilized a relatively large number of distinct video stimuli, making it improbable that participants were able to memorize specific stimuli, and because the order of stimulus presentation was randomized and counterbalanced across participants.

We encourage future investigators to conduct experiments under increasingly more realistic conditions, to further strengthen the congruence between experimental findings and the real world, as well as implementing binocular eye-tracking for more detailed gaze information. Moreover, more comprehensive recruitment of expert individuals to allow a differentiation on a player position basis, as well as the incorporation of dedicated AOI analyses, would result in a more fine-grained investigation of expertise-related perceptual-cognitive dynamics.

## Conclusion

In conclusion, lower rates of fixations, as well as longer fixation durations were found in relation to expertise during the observation of dynamic volleyball video sequences. The preceding phase, in which players' preparation before the start of a rally was observed, revealed a different pattern, indicating more exploratory gaze behavior at an earlier stage of information processing from experts. We expanded on previous findings by investigating temporal anticipatory dynamics through distinct time intervals during scene observation, as well as by incorporating a control condition to gain insight

into domain-specific task effects on gaze behavior. We propose that future research should put further emphasis on the temporal aspect of perceptual-cognitive processing, as well as investigate how experimental findings are transferable to real-world scenarios.

## Supporting information

**S1 Data. Survey data.** File including survey data.
(CSV)

**S2 Data. Fixation data.** File including eye-tracking fixation data.
(CSV)

**S3 Data. Saccade data.** File including eye-tracking saccade data.
(CSV)

**S4 Data. Missing data.** File including information on eye-tracking missing data.
(CSV)

**S5 Code. Analysis code.** R code for statistical analyses.
(R)

**S6 Table. Statistical tables.** File including tables from complementary statistical analyses.
(PDF)

## Acknowledgments

The authors would like to thank the Field of Excellence COLIBRI (Complexity of Life in Basic Research and Innovation, University of Graz).

## Author contributions

**Conceptualization:** Thomas Kanatschnig, Norbert Schrapf, Otto Lappi, Christof Körner, Markus Tilp, Silvia Erika Kober.

**Data curation:** Thomas Kanatschnig.

**Formal analysis:** Thomas Kanatschnig, Živa Korda.

**Investigation:** Thomas Kanatschnig, Lisa Leitner.

**Methodology:** Thomas Kanatschnig, Živa Korda.

**Project administration:** Thomas Kanatschnig, Silvia Erika Kober.

**Resources:** Norbert Schrapf, Christof Körner.

**Software:** Christoph Anzengruber.

**Supervision:** Markus Tilp, Silvia Erika Kober.

**Validation:** Thomas Kanatschnig.

**Visualization:** Thomas Kanatschnig.

**Writing – original draft:** Thomas Kanatschnig.

**Writing – review & editing:** Thomas Kanatschnig, Živa Korda, Otto Lappi, Christof Körner, Markus Tilp, Silvia Erika Kober.

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
