## [Decision Letter · Decision Letter 0]

31 Mar 2025

PONE-D-25-10199More efficient gaze behavior in relation to expertise during tactical decision-making in volleyballPLOS ONE

Dear Dr. Kanatschnig,

Thank you for submitting your manuscript to PLOS ONE. After careful consideration, we feel that it has merit but does not fully meet PLOS ONE’s publication criteria as it currently stands. Therefore, we invite you to submit a revised version of the manuscript that addresses the points raised during the review process.

Specifically, the reviewers highlighted some significant methodological shortcomings which I feel need to be addressed first before entering the manuscript into subsequent rounds of the review process. 

We look forward to receiving your revised manuscript.

Kind regards,

Job Fransen

Academic Editor

PLOS ONE

“This study was supported by the Field of Excellence COLIBRI (Complexity of Life in Basic Research and Innovation, University of Graz).”

“This study was financially supported by the University of Graz for coverage of publication fees. No additional external funding was received for this study.”

4. We note that Figures 1 and 2 in your submission contain copyrighted images. All PLOS content is published under the Creative Commons Attribution License (CC BY 4.0), which means that the manuscript, images, and Supporting Information files will be freely available online, and any third party is permitted to access, download, copy, distribute, and use these materials in any way, even commercially, with proper attribution. For more information, see our copyright guidelines: http://journals.plos.org/plosone/s/licenses-and-copyright.

a. You may seek permission from the original copyright holder of Figures 1 and 2 to publish the content specifically under the CC BY 4.0 license.

5. We note that there is identifying data in the Supporting Information file <S1_Data.csv>. Due to the inclusion of these potentially identifying data, we have removed this file from your file inventory. Prior to sharing human research participant data, authors should consult with an ethics committee to ensure data are shared in accordance with participant consent and all applicable local laws.

-Location data

Please remove or anonymize all personal information, ensure that the data shared are in accordance with participant consent, and re-upload a fully anonymized data set. Please note that spreadsheet columns with personal information must be removed and not hidden as all hidden columns will appear in the published file.

Additional Editor Comments:

Dear authors

The reviewers have suggested significant revisions to your paper before it can be considered further. I specifically recommend you address the methodological issues highlighted by both thoroughly, as for me they still offer a consideration of whether your work should be sent out to another round of reviews.

The work you submitted is undoubtely interesting and I look forward to seeing your responses.

Reviewers' comments:

Reviewer's Responses to Questions

**Comments to the Author**

1. Is the manuscript technically sound, and do the data support the conclusions?

Reviewer #1: Partly

Reviewer #2: Partly

2. Has the statistical analysis been performed appropriately and rigorously? 

Reviewer #1: Yes

Reviewer #2: Yes

3. Have the authors made all data underlying the findings in their manuscript fully available?

Reviewer #1: Yes

Reviewer #2: Yes

4. Is the manuscript presented in an intelligible fashion and written in standard English?

Reviewer #1: Yes

Reviewer #2: Yes

5. Review Comments to the Author

Reviewer #1: General comments:

The manuscript addresses a relevant and interesting topic related to perceptual-cognitive expertise in sport, focusing on gaze behavior in volleyball players of varying expertise levels. The use of a dynamic, domain-specific task and the inclusion of a temporally structured analysis (Preparation vs. Rally phases) are clear strengths of the study. The methodological approach, including eye-tracking and linear mixed effects modeling, is appropriate and generally well-described.

At the same time, several aspects of the study require clarification, expansion, or revision to strengthen its scientific contribution and improve methodological transparency. The following points outline the main areas where the manuscript could benefit from improvement.

Major comments:

1. Lack of an explicitly stated hypothesis

• While the introduction is coherent and based on relevant literature, it does not culminate in a clearly formulated research hypothesis or set of hypotheses. A testable hypothesis—derived from prior findings—would help frame the aims of the study and guide interpretation of the results, especially with regard to expected differences across groups, tasks, and phases.

2. Justification for the novelty of the study

• The topic of perceptual-cognitive expertise in volleyball is not new, and several previous studies have addressed gaze dynamics in this domain (e.g., Fortin-Guichard et al., 2020; Piras et al., 2014). The authors briefly mention some distinguishing aspects (e.g., inclusion of an intermediate group, temporal analysis), but a more explicit and focused explanation of what new insights this study provides would enhance the manuscript’s originality and relevance.

3. Methods section:

Unequal group sizes and sample power

• The three expertise groups are notably unbalanced (n = 14 experts, n = 25 amateurs, n = 19 novices). This raises concerns regarding the statistical power of the analysis, particularly for detecting interaction effects in the linear mixed models.

• The manuscript does not report any a priori power analysis. The authors are encouraged to justify the sample size based on expected effect sizes or acknowledge this limitation explicitly in the discussion.

Monocular eye tracking (right eye only)

• The use of monocular (right-eye) tracking is not uncommon in eye-tracking research. However, given the task’s emphasis on visual attention and the importance of peripheral vision in sport-specific scenarios, it would be helpful for the authors to justify this decision or at least discuss its implications.

Statistical approach – appropriateness and clarity

• The use of linear mixed-effects models is appropriate for the study design. Nonetheless, the manuscript would benefit from:

• A clearer rationale for the choice of random effects structure,

• A statement on whether model assumptions (e.g., normality, homoscedasticity) were checked and met,

• More detail on how task block order was counterbalanced across participants.

4. Results: Missing analysis of prediction accuracy

• The main task required participants to anticipate the setter’s pass—a decision-making component that is central to the study’s rationale. However, the manuscript does not report participants’ accuracy in the Prediction condition. Including this data would provide essential behavioral context and allow stronger inferences about perceptual efficiency in relation to expertise.

5. Limitations section is too brief

• The limitations section should be expanded to reflect important considerations that may affect the interpretation of results. In particular:

• The lack of analysis by player position, which is known to influence visual search behavior (Fortin-Guichard et al., 2020),

• The possibility of learning or repetition effects, as participants viewed the same stimuli in both task conditions,

• The use of monocular tracking and constrained head position, which may reduce ecological validity in a sport-performance context.

Minor suggestions:

• Consider defining the concepts of “gaze anchors” or “visual pivots” more precisely and referencing foundational sources in this area.

• Lines 226–227 (Methods): the criteria for classifying saccades (velocity > 30°/s, etc.) should be supported with methodological references.

Reviewer #2: The authors present a study where they employed the temporal occlusion paradigm on video stimuli of national-level volleyball games (recorded from a bird's eye view, probably with a camera at the stand). The videos showed the scene from the preparation of a service "until the point before the setter plays the pass". In a (blocked, counter-balanced) “Prediction” condition, participants of three different expertise levels (19 novices, 25 amateur and 14 expert players) had to select one of four potential destinations of the setter’s pass. In a second “Control” condition, the task of the participants was simply to select one of three potential locations of the service player, a task without domain-specific knowledge requirements with respect to prediction of outcomes. Multiple gaze metrics (with respect to fixations and saccades) were recorded and analysed for the two time intervals before and after the service. In line with previous research, the authors found expertise-dependent differences for all groups (fewer fixations, longer fixation duration, fewer saccades), but only in the later time interval. In the first interval, only novices differed with comparatively higher rates of fixations and saccades, potentially indicating a more exporative gaze strategy of experts to gather situational knowledge and understanding already before the critical phase. The authors conclude that these findings emphasize the "importance of temporal dynamics" as well as a "comprehensive operationalization of perceptual-cognitive processes related to expertise".

While the authors seem follow a comprehensive and rigorous (statistical) analysis procedure, to me, the study design itself has several weaknesses, of which some seem not resolvable. Maybe this is due to the fact that the eye tracking part was only planned as "a secondary analysis [...] for the exploratory investigation of viewing behavior,[...] to look at the frequency of eye saccades and fixation times in specific regions of interest" (https://osf.io/exhr3). In the following, I will comment on four of theses weaknesses.

First and foremost, the authors claim a "high ecological validity of the stimulus" (e.g., l. 469). However, the bird's eye view is obviously a very unnatural perspective for a player. While this perspective might be ok to assess coaches (for what the original material was intended as far as I understood), there is a substantial amount of evidence that perceptual expertise unfolds (best) in representative environments and task conditions. In the study at hand, the perspective on the visual information, the purely passive reception (even with constrained head position), but also the task (to predict the ball's final destination or to judge the position of the service player) and the way responses had to be given (by button presses) seem not to align with consequences arising from this evidence. If one assumes that a player actively engages with the environment (and, e.g., creates and manipulates perception by moving around, but also changes information intake/gaze behavior depending on the task at hand), my main concern is related to what the study results really tell us with respect to expertise differences in volleyball - and my critical answer would be "close to nothing". This becomes even more clear when considering that the gaze metrics analysed (apart from the descriptive gaze density plots) only relate to temporal aspects of gaze behavior independent from the location of the gaze and therefore are not related to the information in the scene but rather a generic aspect of oculomotor behavior when watching a volleyball serve. I am not sure how to restructure the manuscrupt to resolve this issue and to still extend to the currently available body of knowledge with respect to gaze behavior, but I would be open to further discuss ideas to overcome it with the authors.

Second, I have come across some potential issues with respect to the experimental design and analysis (some of which are only mentioned in the already published work (<https: 10.1371=" " doi.org=" " journal.pone.0318234=" ">):

- "At the end of the video, the last frame (immediately before the pass of the setter) was displayed to the participants for 0.5 s as a freeze frame, which was defined as the Freeze phase."  This potentially affects gaze behavior as the temporal availability of the last frame's information is much longer compared to the real-life situation and therefore contrasts the idea of the temporal occlusion paradigm.

- "Near side vs far side": in the original video set have been services from both teams - did you investigate them here as well? If so, I would expect differences in gaze behavior between these conditions due to the different visual angles which are covered.

- What is exactly "the point before the setter plays the pass"? The last frame, before the setter touches the ball or when the ball leaves the hand of the setter? Please be more clear here.

- Could the fixation cross in the screen's center affect the density plot or even the reported results? It seems they already fixate on a (potentially) information richt area and therefore could be biased in their natural gaze behavior. For better control, a fixation on a non information rich area at the edges of the screen seem advisable.

- How are the colors in the density plot scaled? How do differences in the amount of fixations across conditions affect the color / this plot?

- No sample size calculations for eye tracking have been reported/performed. How did you come up with the sample sizes - can you also comment/discuss the varying amount of participants in the different groups?

- Why don't you report gaze behavior differences with respect to unsuccessful/successful trials, but rather only across groups?

Third, there are several mentions of expertise-dependent efficiency differences when extracting information during fixations. To me, these arguments seem purely speculative without taking the relative locations of the fixation into account. So also this line of argumentation needs substantial rework and better justification.

And fourth and finally, also aspects with respect to the use of peripheral information can only properly discussed when analysis which information is visible in which peripheral eccentricity and whether it can be used to inform prediction and/or decision making.

I think at the current stage, more detailed feedback on the manuscript is not needed - to me these four issues need to be resolved first and this will probably cause a major rework of the manuscript's structure.</https:>

6. PLOS authors have the option to publish the peer review history of their article (what does this mean? ). If published, this will include your full peer review and any attached files.

**Do you want your identity to be public for this peer review?** For information about this choice, including consent withdrawal, please see our Privacy Policy .

Reviewer #1: No

Reviewer #2: No

---

## [Author Response · Author response to Decision Letter 1]

15 May 2025

Response to Reviewers

Concerning the revision of the manuscript More efficient gaze behavior in relation to expertise during tactical decision-making in volleyball (PONE-D-25-10199)

Dear Reviewers,

Dear Editors of PLOS ONE,

On behalf of my co-authors and myself, I would like to thank you for your thoughtful and constructive feedback on our manuscript. We hereby present to you our responses to all points that were raised and the subsequent changes we made to the manuscript. Please note that all mentions of specific line numbers (L. xx-xx) in our responses refer to the file “Revised Manuscript with Track Changes”.

With kind regards,

Thomas Kanatschnig

Reviewer 1

General comments:

The manuscript addresses a relevant and interesting topic related to perceptual-cognitive expertise in sport, focusing on gaze behavior in volleyball players of varying expertise levels. The use of a dynamic, domain-specific task and the inclusion of a temporally structured analysis (Preparation vs. Rally phases) are clear strengths of the study. The methodological approach, including eye-tracking and linear mixed effects modeling, is appropriate and generally well-described.

At the same time, several aspects of the study require clarification, expansion, or revision to strengthen its scientific contribution and improve methodological transparency. The following points outline the main areas where the manuscript could benefit from improvement.

We would like to thank you for the detailed and constructive feedback on our manuscript. We deliberated on each point carefully and gave our best effort in tackling each of them thoroughly, as well as providing additional information to further strengthen our message. We hope that our revisions further support your positive impression of our work.

1. Lack of an explicitly stated hypothesis

• While the introduction is coherent and based on relevant literature, it does not culminate in a clearly formulated research hypothesis or set of hypotheses. A testable hypothesis—derived from prior findings—would help frame the aims of the study and guide interpretation of the results, especially with regard to expected differences across groups, tasks, and phases.

We agree and would like to thank you for making us aware of the fact that this aspect needs more information/clarification. First and foremost, we must clarify that the lack of specific testable hypotheses was a deliberate decision. A paragraph was added at the end of the Introduction section, which explains that because the analysis of gaze behavior was planned as a secondary explorative type of investigation for this study (stated as such in the preregistration for the research project, available at https://osf.io/exhr3), we have not defined specific a-priori hypotheses for the present analyses regarding gaze behavior. Instead, our approach was to examine expertise-related effects (novices vs. amateurs vs. experts) on gaze behavior in the framework of our experiment (tactical decision-making) and provide a comprehensive report of our findings for relevant gaze metrics (fixations, saccades). This is the reason why we backed away from formulating any concrete hypotheses for this work, however, for the sake of transparency we now made the explorative nature of this work clearer in order to avoid any misinterpretations regarding our research objectives (L. 103-106).

2. Justification for the novelty of the study

• The topic of perceptual-cognitive expertise in volleyball is not new, and several previous studies have addressed gaze dynamics in this domain (e.g., Fortin-Guichard et al., 2020; Piras et al., 2014). The authors briefly mention some distinguishing aspects (e.g., inclusion of an intermediate group, temporal analysis), but a more explicit and focused explanation of what new insights this study provides would enhance the manuscript’s originality and relevance.

Thank you for this suggestion. We added an explanation about what we deem to be the most important distinguishing factor of our investigation, which is the combination of four key methodological aspects of the experimental design, to provide a comprehensive picture of dynamic perceptual-cognitive processes underlying tactical decision-making in volleyball (L. 118-123).

3. Methods section:

Unequal group sizes and sample power

• The three expertise groups are notably unbalanced (n = 14 experts, n = 25 amateurs, n = 19 novices). This raises concerns regarding the statistical power of the analysis, particularly for detecting interaction effects in the linear mixed models.

We do see that the unequal group sizes can be considered as a limitation of our study. While we were able to reach our minimum targeted sample sizes for the Amateur and Novice groups (see next point regarding power calculation), the recruitment of professional volleyball players for the Expert group revealed to be difficult. Nevertheless, while we would have certainly preferred a more balanced sample, we are confident that our LMM results reflect reliable effects for two key reasons: (i) the high level of data quality we were able to achieve (L. 292-301), and (ii) the consistent patterns we observed across all gaze metrics, which would likely not have emerged had the LMM estimates been substantially biased due to sample imbalance. Moreover, LMMs are, in principle, designed to handle unbalanced data, making them arguably the most appropriate statistical approach for this scenario [1].

• The manuscript does not report any a priori power analysis. The authors are encouraged to justify the sample size based on expected effect sizes or acknowledge this limitation explicitly in the discussion.

Thank you for raising this important point. Regarding statistical power we must clarify, that due to the fact that our study in its core revolved around neurophysiological substrates of the relationship between sport expertise and efficiency (see [2]), our sample size calculations were based on studies investigating expertise-related effects of EEG frequency band power in the theta and alpha range [3]. Our a priori power calculations (see https://osf.io/exhr3) revealed that a minimum of 20-21 participants per group would be necessary to detect the expected effects in the respective EEG bands, which is what we were able to achieve in case of the Novice and Amateur groups (taking EEG data loss into account), but unfortunately fell short for the Expert group. While our power calculations were centered around the EEG analyses, we did find our sample/group sizes to be fairly comparable to those of previous investigations in the domain of perceptual-cognitive investigations relating to expertise. We are certainly far away from sample sizes of 27 [4] or even 62 [5] (here referring solely to the number of professional/expert individuals), yet the majority of the relevant literature that we refer to in our manuscript reports sample sizes of 10-15 individuals in a respective Expert group [6–9], which is why we think that our sample is comparable to these previous investigations with respect to statistical power. Furthermore, given our experimental design, we gathered multiple data points from each participant, which has been shown to improve statistical power [10]. Still, to acknowledge that a dedicated a priori power analysis for effects related to our investigated gaze metrics is lacking, we added a brief discussion of this point to the Limitations section (L. 546-552).

Monocular eye tracking (right eye only)

• The use of monocular (right-eye) tracking is not uncommon in eye-tracking research. However, given the task’s emphasis on visual attention and the importance of peripheral vision in sport-specific scenarios, it would be helpful for the authors to justify this decision or at least discuss its implications.

We agree that the use of binocular tracking could have provided more detailed information regarding depth perception and peripheral vision. The decision to use monocular tracking was made primarily for practical reasons, given its simpler calibration procedure, which aligned better with our co-registration design with EEG, as well as the deliberation that our primary gaze metrics of interest could be reliably captured using monocular tracking. Nevertheless, we acknowledge your comment and have now added this point to the Limitations section (L. 561-565).

Statistical approach – appropriateness and clarity

• The use of linear mixed-effects models is appropriate for the study design. Nonetheless, the manuscript would benefit from:

• A clearer rationale for the choice of random effects structure,

A more detailed explanation concerning the random effects structure of our models has now been added to the manuscript (L. 317-328).

• A statement on whether model assumptions (e.g., normality, homoscedasticity) were checked and met,

Model diagnostics have initially already been calculated to check for potential deviations from model assumptions and are reproducible alongside our main analyses by means of our provided analysis data and R code under “Supporting information”. Multicollinearity was generally found to be no concern, as generalized variance inflation factors [GVIF^(1/(2*Df))] of all fixed effects on each DV revealed values in an acceptable range [11]. Normality of residuals was checked on each level combination of Group, Task and Phase by inspection of skewness and kurtosis values, histograms, as well as Q-Q plots, which revealed deviations in a few instances, yet not to an extreme extent. Plotting residuals against fitted values to inspect potential issues with heteroscedasticity did not reveal any potential bias as scatter clouds generally followed a circular/oval pattern. Overall, we acknowledge that our data does not fit model assumptions perfectly in regard to normality, which is why we now included a sentence to the manuscript making aware of this circumstance (L. 335-337). However, the deviations we identified did not yield strong reasons for concern regarding the reliability of our estimations. Moreover, LMMs have been found to be quite robust to violations of assumptions, which according to Schielzeth et al. [1] “…should allow researchers to use mixed-effects models even if the distributional assumptions are objectively violated.”

• More detail on how task block order was counterbalanced across participants.

For the purposes of this response and the revisions made to the manuscript, we refer to the experimental design descriptions detailed in our previous publication Kanatschnig et al. [2]. Relevant sections from that publication will be quoted verbatim below. In the Materials section of our previous publication it is stated:

“…, the task was divided into two separate blocks. In the Near-side block the actions of interest (i.e., the setting in the Prediction condition and the service in the Control condition) occurred on the nearer side of the court (court side in front of the net), whereas in the Far-side block the actions occurred on the farther side of the court (court side behind the net), with respect to the camera’s field of view of the video recordings. The number of videos for the Prediction condition was 25 in the Near-side and 38 in the Far-side block. The videos of the Prediction condition in one block functioned as videos for the Control condition in the other block, given that the service always happened on the opposite court side. …”

Further, in the Design and general procedure section of our previous publication it is stated:

“… Before the first task block, participants carried out a practice run with different stimuli to get familiar with the task. Participants completed the first block, which was the Near-side block, in which the order of Prediction and Control conditions was counterbalanced across participants. After completing the Near-side block, participants carried out another training run for the second block, which was the Far-side block. During the Far-side block the order of Prediction and Control conditions was again counterbalanced. …”

In the first citation we describe the existence of a factor “Court” which describes on which side of the court (i.e., Near-side vs. Far-side) the respective action of interest depending on the task condition at hand (i.e., Prediction vs. Control) within each video occurred. This factor Court could however not be meaningfully investigated for reasons we discuss within the expanded description of our LMM random effect structure in the revised manuscript (L. 317-328), and it has therefore been omitted in the present investigation of gaze behavior. In the second citation we describe what the core of your comment refers to, namely the task block order and counterbalancing. It is stated here that there were two separate presentation blocks, Near-side and Far-side, referring to which court side the actions of interest occurred in the videos. All 63 videos occurred exactly one time per presentation block, only the task connected to the video changed depending on the respective block. Within each block the order of task conditions was counterbalanced across participants, i.e., one participant started the Near-side block with the Prediction condition, followed by the Control condition and then started the Far-side block again with Prediction, followed by Control, while another participant started each block with the Control condition followed by the Prediction condition. We now expanded the description of task block order and counterbalancing to provide a more detailed explanation regarding our procedure (L. 228-242); however, we did not include information regarding the omitted factor Court (Near-side vs. Far-side) from our previous publication [2], as we felt that it does not serve any additional information necessary to understand the basic principles of our present investigation and could therefore potentially distract the reader from more relevant information.

4. Results: Missing analysis of prediction accuracy

• The main task required participants to anticipate the setter’s pass—a decision-making component that is central to the study’s rationale. However, the manuscript does not report participants’ accuracy in the Prediction condition. Including this data would provide essential behavioral context and allow stronger inferences about perceptual efficiency in relation to expertise.

Thank you for this suggestion. We fully agree and have now added additional information regarding the behavioral results of the experiment, which (as stated in the manuscript) have been reported previously by Kanatschnig et al. [2] alongside the EEG analyses (L. 506-511).

5. Limitations section is too brief

• The limitations section should be expanded to reflect important considerations that may affect the interpretation of results. In particular:

• The lack of analysis by player position, which is known to influence visual search behavior (Fortin-Guichard et al., 2020),

• The possibility of learning or repetition effects, as participants viewed the same stimuli in both task conditions,

• The use of monocular tracking and constrained head position, which may reduce ecological validity in a sport-performance context.

Thank you again for these thoughtful observations. We have added justifications/discussions for each of them to the Limitations section (L. 539-580).

Minor suggestions:

• Consider defining the concepts of “gaze anchors” or “visual pivots” more precisely and referencing foundational sources in this area.

• Lines 226–227 (Methods): the criteria for classifying saccades (velocity > 30°/s, etc.) should be supported with methodological references.

These methodological aspects have now been adapted in the manuscript as well (L. 59-64; 276-278).

Reviewer 2

The authors present a study where they employed the temporal occlusion paradigm on video stimuli of national-level volleyball games (recorded from a bird's eye view, probably with a camera at the stand). The videos showed the scene from the preparation of a service "until the point before the setter plays the pass". In a (blocked, counter-balanced) “Prediction” condition, participants of three different expertise levels (19 novice

---

## [Decision Letter · Decision Letter 1]

30 Jul 2025

PONE-D-25-10199R1More efficient gaze behavior in relation to expertise during tactical decision-making in volleyballPLOS ONE

Dear Dr. Kanatschnig,

Thank you for submitting your manuscript to PLOS ONE. After careful consideration, we feel that it has merit but does not fully meet PLOS ONE’s publication criteria as it currently stands. Therefore, we invite you to submit a revised version of the manuscript that addresses the points raised during the review process.

 There were conficting appraisals from reviewers. A third reviewer chose not to reject the article, and offered some methodological considerations which need to be addressed. Hence, when the concerns of reviewer two and three are addressed, I will make a reasonable decision as to whether they were suitably addressed and the manuscript can be accepted for publication. 

We look forward to receiving your revised manuscript.

Kind regards,

Job Fransen

Academic Editor

PLOS ONE

Journal Requirements:

Additional Editor Comments:

Dear authors

As you know, we have had some issues in terms of conflicting reviews from the reviewers. I understand this article has been in review for a while, but I would rather have an article that is published and fully scrutinised, than one we have glossed over. The third reviewer has provided some additional revisions, and I would like to see you review their comments and those of reviewer two (who rejected). I will then make a final call on the suitability of the article for publication.

Again my apologies, but there is no substitute for a good and thorough review process, and I am very grateful to the reviewers for their comments.

Reviewers' comments:

Reviewer's Responses to Questions

**Comments to the Author**

1. If the authors have adequately addressed your comments raised in a previous round of review and you feel that this manuscript is now acceptable for publication, you may indicate that here to bypass the “Comments to the Author” section, enter your conflict of interest statement in the “Confidential to Editor” section, and submit your "Accept" recommendation.

Reviewer #1: (No Response)

Reviewer #2: (No Response)

Reviewer #3: (No Response)

2. Is the manuscript technically sound, and do the data support the conclusions?

Reviewer #1: Yes

Reviewer #2: No

Reviewer #3: Yes

3. Has the statistical analysis been performed appropriately and rigorously? 

Reviewer #1: Yes

Reviewer #2: I Don't Know

Reviewer #3: Yes

4. Have the authors made all data underlying the findings in their manuscript fully available?

Reviewer #1: Yes

Reviewer #2: Yes

Reviewer #3: Yes

5. Is the manuscript presented in an intelligible fashion and written in standard English?

Reviewer #1: Yes

Reviewer #2: Yes

Reviewer #3: Yes

6. Review Comments to the Author

Reviewer #1: The authors addressed most of the comments and implemented the changes that were feasible. I appreciate the effort they put into the revision. I have no further comments.

Reviewer #2: I pointed out 4 key issues in my previous review, which did render more detailed feedback on the manuscript superfluous at that stage. While the authors tried to address these in their revised work and the rebuttal, I am still not convinced that the submitted work provides valuable insights. I will try to be more concise this time and add details with respect to the content rather than the methodological aspects only.

l. 39f. You call your task a “tactical decision-making task”. I would rather argue that the task is an “observation and prediction task”. This notion is important as tactical decision making implies that the participant is engaged in the situation and tries to solve it, while in your setting the person is only observing movement patterns (related to my previous review: in unnatural viewing conditions for players) and trying to predict the setter’s target player.

l.59f. Gaze anchors and visual pivots are only meaningful if analysed with respect to the underlying SPATIOtemporal aspects of the stimuli. A purely temporal analysis cannot unveil these aspects (it could be, that more experienced persons just “know” where to look at and saccade directly to that stimulus (as you pointed out in l.68f), which would directly result in lower amounts of saccades and potentially longer fixations). This already calls for an AOI-based analysis of your data – which you already pointed out in your preregistration (“For the exploratory investigation of viewing behavior, we plan to look at the frequency of eye saccades and fixation times in specific regions of interest.”) and inferred its necessity in a later section of your introduction (l.88f). The narrowing down to fixation and saccade rates (l.95f.) is not substantiated at all.

l.64f. How does Faubert’s 3D-MOT task fit into the line of argumentation here?

l.101f. This is the first mention of the control task. To me, this control task is just a completely different task with clearly differing visual demands (e.g. focusing on one player only vs. a “tactical situation”). The rationale for incorporating this task does not become clear. To me, a task which purely manipulates domain knowledge requirements while keeping perceptual requirements identical would be the only fair comparison. Moreover, if employing standardized lab-based experiments, you should respect the max-con-min principle; it is not clear at all why you alter the amount of response options between experimental and control task (3 vs 4, see also l. 193f.). Is it correct that the videos also differed between the experimental and control task for the same condition as you defined the “action of interest” to be on the near or far side of the net (l. 231f.)? If so, this additionally confounds your experimental manipulation.

l.103f. This section is important but for me way too generic to be meaningful. A good introduction should set the stage, follow a funnel-like structure (narrowing down) and result in a clear-cut research question (even if you performed an exploratory design).

l. 107f. Parts of this section seem to belong to the introduction.

l. 166f. You showed women’s indoor volleyball situations to males and females but you did not mention this potentially confounding aspect. It is expected that domain-specific expertise unfolds (best) when confronted with situations the participants have experienced a lot – which is not true for men observing women’s games. This argument again can be extended with respect to the bird’s eye perspective (and even from the “wrong side” when confronted with the “FAR” condition) – but I raised this point during the first phase of the review already. At least a discussion with respect to higher (and unnatural) task demands (e.g., as you need to mentally rotate) compared to the real situation and their effects on your measures seems necessary.

l. 177f. I do not find a discussion of the varying duration of the preparation phase – from 3 to 10 seconds is quite a range which might influence gaze behavior (e.g., triggering exploratory viewing behavior purely because of longer durations).

l. 179f. It is not clear how a natural game situation can be “fixed” to exactly 3 seconds. There must be some time variation from service to the ball contact of the setter.

l. 247f. You argued in your rebuttal that the 500ms static image will not influence the viewing behaviour during the preparation and rally phase. I strongly disagree here as experts are very efficient in extracting relevant information (as you pointed out in your introduction). So, within a couple of trials I expect that experts only focus on extracting the information in the last 500ms – maybe with an exploratory viewing behaviour to optimally locate the gaze to the most information rich area. This would mean that the observed gaze behaviour difference during rally is completely irrelevant for them to solve the task. This is a fundamental flaw of the experiment, and you can only get (exploratory) insights when doing spatiotemporal analyses. This aspect has also consequences for your discussion (l. 454f) rendering these insights potentially meaningless for solving the task.

l.266f. There was a large initiative in establishing a reporting guideline for eyetracking experiments (Dunn et al. 2023, https://doi.org/10.3758/s13428-023-02187-1), which tries to make eye tracking experiments more replicable and understandable for other researchers. This guideline should be followed. Moreover, the reference (Walcher et al. [36]) seems not to employ a standard way of classifying gaze into fixations and saccades – which contributes to confusion with respect to terminology (a fixation is something different than a smooth pursuit eye movement).

l. 321f. Your rationale for constructing the model emphasizes the point I already made in my prior review that viewing conditions are critical and systematically affect the gaze behavior observed. In my view, this is very critical. Turning it to the positive side: Maybe one could use this (unintended) manipulation of stimulus demands (having information in smaller or larger ranges of visual angles) to explore gaze behavior differences between near and far situations. The current way of analysis, however, weakens the findings.

l.337f. Can you be clearer about the effect of employing Benjamini-Hochberg adjustments to your stats?

l.360f. Figure heading: the coloring of the graph still seems misleading as one cannot compare between conditions (i.e. the same lightness means different things due to varying amounts of data). I think you should standardize this to help the reader understand and compare the graphs. In addition, these graphs only take on condition and neglect the other – so I guess it would be informative to see gaze densities but maybe also scan-paths for all conditions (to get a full picture of the viewing pattern).

l. 387f./401f. Please refrain from using “fixation rate” and “fixation duration” for the time intervals in which the participants can extract information from the visual scene – this is something different!

To sum up, due to the points raised in my previous review and now here in more detail I do not see the authors’ interpretations and conclusions justified. Maybe the data set might still provide interesting hints for future research when analysed in a clever (e.g., turning experimental design flaws into interesting aspects) and spatiotemporal (including AOI-based, but also more fine grained temporal analyses (e.g., separating the last 500ms from the rest of the rally)) way.

Reviewer #3: The paper reports on an eye tracking experiment conducted with novice, amateur and expert volleyball players. Although not presenting entirely novel findings, this was acknowledged by the author, and the paper still adds some insights to the known literature, as well as confirming some previous findings.

The authors already replied to the comments of two reviewers, which improved the manuscript. I would like to see this article published, but I think there needs to be some further clarification first. Below some additional comments to those already made by the other reviewers.

Comment #1

The title does not suit the content of the paper. The task in the experiment is to predict the direction of the pass. This is not tactical decision-making, it is pattern recognition and (mostly) anticipation. The participants do not need to make a decision (e.g. what is the best course of action), they need to predict. So I believe it is not suitable to refer to this experiment as decision making nor as tactical.

Comment #2:

Please refrain from using the wording ‘fewer fixations of longer duration’, as this could mean both that experts made fewer long fixations, or that they made fewer fixations but they were of longer duration. This also occurs a couple of times in the discussion.

See: Vansteenkiste, P. (2022). Fewer fixations of longer duration can lead to more fixations of longer duration: A commentary on the description of the visual behaviour of expert performers in sports. German Journal of Exercise and Sport Research, 52(1), 198-199.

Comment #3:

It is not clear to me why both Fixation rate and Saccade frequency is included in this study. Shouldn’t fixation rate and saccade rate be the same, assuming that a fixation is always followed by a saccade? In what case could fixation rate and saccade frequency be different? Unless I’m mistaking, I believe this metric (saccade frequency) could (and should) be omitted from the paper.

Comment#4:

With multiple conditions (group, task, phase) it is sometimes difficult to follow the results. The table refers to ‘task’ but the text refers to ‘condition’ when referring to ‘Prediction’ or ‘Control’. I would suggest to keep using the same term to refer to either the task or the condition.

In line with this, it is also not easy to compare the actual results to each other (e.g. fixation duration of experts an novices in the prediction task during the rally phase). Although this information is in the text, I think an extra table (perhaps as supplementary info) would benefit the clarity of the results. I would suggest to move table 3 to appendix and replace it by a similar table with the average +- SD results.

Comment #5

The experts in the prediction condition during rally had a Fixation duration of 1041 ms. However, their Fixation rate in this phase and condition was 1.21 fix/sec. How is an average frequency of more than 1 fixation per second possible if the average fixation duration is more than 1 second long?

Comment #6:

The authors already added the results of accuracy (from another paper: Kanatschnig et al. [2]) to the discussion, but I believe that it would be better to report this as results as well. However, I agree with the authors that this should not be elaborated on as it its discussed in another paper.

Comment #7:

L540: “It is necessary to emphasize that we were only able to observe these dynamic 541 patterns results through the distinction between Preparation and Rally phases, but also through 542 the inclusion of amateurs as an intermediate level group in our experimental design.”

Missing the word ‘not’: … were not only able to observe … ?

Comment #8:

The authors should also refer to a recent study with a very similar experimental design:

Zhu, R., Zou, D., Wang, K., & Cao, C. (2024). Expert performance in action anticipation: visual search behavior in volleyball spiking defense from different viewing perspectives. Behavioral Sciences, 14(3), 163.

7. PLOS authors have the option to publish the peer review history of their article (what does this mean? ). If published, this will include your full peer review and any attached files.

**Do you want your identity to be public for this peer review?** For information about this choice, including consent withdrawal, please see our Privacy Policy .

Reviewer #1: No

Reviewer #2: No

Reviewer #3: **Yes: ** Pieter Vansteenkiste

---

## [Author Response · Author response to Decision Letter 2]

22 Sep 2025

Response to Reviewers

Concerning the revision of the manuscript: More efficient gaze behavior in relation to expertise during tactical decision-making in volleyball (PONE-D-25-10199R1)

Dear Reviewers,

Dear Editors of PLOS ONE,

On behalf of my co-authors and myself, I would like to thank you again for your valuable feedback on our manuscript. We hereby present to you our responses to all points that were raised and the subsequent changes we performed on the manuscript in the second round of revision. Please note that all mentions of specific line numbers (L. xx-xx) in our responses refer to “Revised Manuscript with Track Changes” (DOCX/PDF).

With kind regards,

Thomas Kanatschnig

Reviewer 1

The authors addressed most of the comments and implemented the changes that were feasible. I appreciate the effort they put into the revision. I have no further comments.

We thank Reviewer 1 for their positive evaluation and insightful feedback on our manuscript.

Reviewer 2

I pointed out 4 key issues in my previous review, which did render more detailed feedback on the manuscript superfluous at that stage. While the authors tried to address these in their revised work and the rebuttal, I am still not convinced that the submitted work provides valuable insights. I will try to be more concise this time and add details with respect to the content rather than the methodological aspects only.

We sincerely thank you, Reviewer 2, for your thoughtful and detailed review of our manuscript. We greatly appreciate the time you took to assess our work. In the following, we aim to address the additional points you raised, clarifying the scope and intention of our work, and outlining why we believe it contributes valuable insights to the literature.

l. 39f. You call your task a “tactical decision-making task”. I would rather argue that the task is an “observation and prediction task”. This notion is important as tactical decision making implies that the participant is engaged in the situation and tries to solve it, while in your setting the person is only observing movement patterns (related to my previous review: in unnatural viewing conditions for players) and trying to predict the setter’s target player.

Thank you for this observation. We agree that, since participants were not able to make an active tactical decision in the experimental situation, the term prediction provides a more accurate description. We have revised the task description throughout the manuscript, replacing “tactical decision-making” with “volleyball anticipation”. We kept the two task condition labels “Prediction” and “Control”, as the term prediction is already in use there for describing our main task condition.

l.59f. Gaze anchors and visual pivots are only meaningful if analysed with respect to the underlying SPATIOtemporal aspects of the stimuli. A purely temporal analysis cannot unveil these aspects (it could be, that more experienced persons just “know” where to look at and saccade directly to that stimulus (as you pointed out in l.68f), which would directly result in lower amounts of saccades and potentially longer fixations). This already calls for an AOI-based analysis of your data – which you already pointed out in your preregistration (“For the exploratory investigation of viewing behavior, we plan to look at the frequency of eye saccades and fixation times in specific regions of interest.”) and inferred its necessity in a later section of your introduction (l.88f). The narrowing down to fixation and saccade rates (l.95f.) is not substantiated at all.

We appreciate your thoughtful evaluation and fully agree that incorporating spatial aspects of gaze behavior could provide an even clearer picture of the perceptual-cognitive processes during setting prediction. As you noted, our preregistration initially included plans for an area-of-interest (AOI) analysis. However, during data inspection we realized that the camera angle – chosen to capture the entire playing field and players to provide as much information possible for setting prediction – did not allow for reliable AOI coding. In many sequences, players were partially or fully occluded by other players, especially during the Rally phase. Gaze fixations could therefore not be uniquely attributed to specific players or interest areas. Given that, we decided that a spatial analysis would risk introducing misleading conclusions. Instead, we focused specifically on the temporal aspect of gaze behavior, where our design allowed us to contribute new insights, e.g., by comparing three expertise levels, including a control condition, and using real-life match recordings as stimuli. In doing so we both replicated prior findings and extended them with previously underexplored aspects of temporal gaze dynamics. We have now expanded this part in the Limitations section (L. 599-604) to explicitly address the absence of a spatial analysis and to clarify why we concluded that it was not feasible in the present study. We hope this demonstrates our awareness of the issue and our intention to inform future work in this direction.

l.64f. How does Faubert’s 3D-MOT task fit into the line of argumentation here?

Our reference to the 3D-MOT task was meant to signify the superiority of professional athletes in adapting to learn and perform during dynamic visual scene observation tasks. Certain core features of the 3D-MOT, such as the necessity of: “… distributing attention among a number of moving targets among distractors… “ [1] are very similar to our task, in which it is likewise necessary to distribute the attention among multiple targets, i.e., the players in our scenario. Therefore, we saw the connection fitting, albeit we see that we did not communicate this point well enough. We have now extended this segment in the manuscript (L. 68-71).

l.101f. This is the first mention of the control task. To me, this control task is just a completely different task with clearly differing visual demands (e.g. focusing on one player only vs. a “tactical situation”). The rationale for incorporating this task does not become clear. To me, a task which purely manipulates domain knowledge requirements while keeping perceptual requirements identical would be the only fair comparison. Moreover, if employing standardized lab-based experiments, you should respect the max-con-min principle; it is not clear at all why you alter the amount of response options between experimental and control task (3 vs 4, see also l. 193f.). Is it correct that the videos also differed between the experimental and control task for the same condition as you defined the “action of interest” to be on the near or far side of the net (l. 231f.)? If so, this additionally confounds your experimental manipulation.

We gladly want to clarify the rationale for the Control condition. In the revised manuscript we now provide a more detailed description of this condition (L. 105–109). Following the definition you provided, it was also our intention to design a control task that manipulates domain knowledge requirements while keeping perceptual demands identical. To this end, we presented the same videos in both conditions, varying only the action of interest: participants predicted the setting outcome in the Prediction condition (domain knowledge beneficial) and identified the service location in the Control condition (domain knowledge not required). We acknowledge that the difference in response options (3 vs. 4) introduces some variance between the two conditions; however, because gaze behavior was measured only during scene observation, i.e., Preparation and Rally phase, and not during the Response phase, we consider this unlikely to have affected our main gaze-related dependent variables. Regarding the number of videos showing the respective actions of interest on each court side (i.e., Near-side and Far-side), in the Prediction condition there were 25 videos in which the setting happened on the Near-side and 38 videos in which it happened on the Far-side of the court, meaning that there is an uneven number of instances where the respective actions of interest (i.e., setting and service) happened on the respective court sides. While ideally the number of videos would have been balanced, this affected all participants equally. Also, to test whether the numbers of videos per condition were significantly different from a uniform distribution, a chi-square test was calculated which yielded a non-significant result (χ2(1) = 2.68, p = .102), implying that a potential confounding effect of the variance in video counts might only be marginal.

l.103f. This section is important but for me way too generic to be meaningful. A good introduction should set the stage, follow a funnel-like structure (narrowing down) and result in a clear-cut research question (even if you performed an exploratory design).

l. 107f. Parts of this section seem to belong to the introduction.

We have now revised these sections (L. 100-138).

l. 166f. You showed women’s indoor volleyball situations to males and females but you did not mention this potentially confounding aspect. It is expected that domain-specific expertise unfolds (best) when confronted with situations the participants have experienced a lot – which is not true for men observing women’s games. This argument again can be extended with respect to the bird’s eye perspective (and even from the “wrong side” when confronted with the “FAR” condition) – but I raised this point during the first phase of the review already. At least a discussion with respect to higher (and unnatural) task demands (e.g., as you need to mentally rotate) compared to the real situation and their effects on your measures seems necessary.

Thank you for reiterating these important points. We have now incorporated this into the Limitations section (L. 604-610).

l. 177f. I do not find a discussion of the varying duration of the preparation phase – from 3 to 10 seconds is quite a range which might influence gaze behavior (e.g., triggering exploratory viewing behavior purely because of longer durations).

A discussion of this aspect has now been added to the manuscript (L. 195-197).

l. 179f. It is not clear how a natural game situation can be “fixed” to exactly 3 seconds. There must be some time variation from service to the ball contact of the setter.

We recognize that this point required adjustment. It is correct that there is a certain degree of variation in the lengths of the rallies. We have now rephrased this section to make this circumstance clearer (L. 200-202). Our videos show matches between women's teams from the same league where rallies are very homogeneous, i.e., very similar in terms of playing speed, ball height, and ball trajectory/distance. The 3 second time interval, therefore, very accurately covered the rally in each respective video and no substantial distortion of the results due to the fixed time interval was expected.

l. 247f. You argued in your rebuttal that the 500ms static image will not influence the viewing behaviour during the preparation and rally phase. I strongly disagree here as experts are very efficient in extracting relevant information (as you pointed out in your introduction). So, within a couple of trials I expect that experts only focus on extracting the information in the last 500ms – maybe with an exploratory viewing behaviour to optimally locate the gaze to the most information rich area. This would mean that the observed gaze behaviour difference during rally is completely irrelevant for them to solve the task. This is a fundamental flaw of the experiment, and you can only get (exploratory) insights when doing spatiotemporal analyses. This aspect has also consequences for your discussion (l. 454f) rendering these insights potentially meaningless for solving the task.

Thank you for raising this important concern. We agree that the Freeze phase could, in principle, provide relevant information, which is why we initially examined it. However, given its short duration (500 ms), almost no unique fixations occurred in this phase, which is why we did not report it in the manuscript. To demonstrate this, we extended our main analysis file “S5_Code”, which now includes additional analysis of the Freeze phase. The Freeze phase analysis of fixation count (not rate, as this phase was only 500ms) shows a similar picture to that of the Rally phase. However, the interpretability of these results is impaired, as mentioned above, due to the short duration of the phase, which resulted in low fixation counts. Furthermore, we saw a floor effect since multiple participants recorded on average only one fixation in this phase, which negatively affected model integrity. To directly address your point regarding the relevance of the Preparation and Rally phases, we now additionally compared gaze behavior between phases within each group. If experts relied exclusively on the Freeze phase, one would expect largely random viewing patterns in the preceding Preparation and Rally phases, with no systematic differences across them. Our newly added pairwise comparisons (added to the main analysis file “S5_Code”), however, show that experts exhibited the strongest differences between Preparation and Rally on our main dependent measures (i.e., fixation rate, fixation duration) in the Prediction condition, with the largest effect sizes of any group. This suggests that their gaze behavior during these phases is not irrelevant but rather reflects an adaptive allocation of attention over the course of video observation. To inspect the results of these additional analyses, please see the file “REVIEW_ONLY_2”. The results can also be reproduced using the available data and analysis code under Supporting information.

l.266f. There was a large initiative in establishing a reporting guideline for eyetracking experiments (Dunn et al. 2023, https://doi.org/10.3758/s13428-023-02187-1), which tries to make eye tracking experiments more replicable and understandable for other researchers. This guideline should be followed. Moreover, the reference (Walcher et al. [36]) seems not to employ a standard way of classifying gaze into fixations and saccades – which contributes to confusion with respect to terminology (a fixation is something different than a smooth pursuit eye movement).

This is very valuable insight. We have revised the Methods section to align with the reporting guidelines of Dunn et al. [2], ensuring that all relevant aspects are now addressed (L. 139–141; 237-320). For the classification of fixations and saccades, we followed a procedure that has been shown to yield reliable results in previous studies [3–5]. Regarding potential confusion about the differentiation between fixations and smooth pursuit movements, we have rephrased the relevant segment for greater clarity (L. 294-298).

l. 321f. Your rationale for constructing the model emphasizes the point I already made in my prior review that viewing conditions are critical and systematically affect the gaze behavior observed. In my view, this is very critical. Turning it to the positive side: Maybe one could use this (unintended) manipulation of stimulus demands (having information in smaller or larger ranges of visual angles) to explore gaze behavior differences between near and far situations. The current way of analysis, however, weakens the findings.

This is an important consideration, and we thank you for raising it. We fully agree that viewing conditions and stimulus demands affect gaze behavior. We constructed a linear mixed-effect model to explicitly capture variance arising from differing visual features of video stimuli, as well as from changing viewing conditions following phase transitions. In the Statistical Analysis section, we clarify relevant considerations regarding statistical implications (L. 335-352).

l.337f. Can you be clearer about the effect of employing Benjamini-Hochberg adjustments to your stats?

Thank you for this suggestion. We expanded the description of the Benjamini-Hochberg adjustment in the Statistical analysis section, clarifying the effect of the Type I error correction method on our pairwise compar

---

## [Decision Letter · Decision Letter 2]

2 Oct 2025

Expertise-driven temporal gaze dynamics during anticipation in volleyball

PONE-D-25-10199R2

Dear Dr. Kanatschnig,

We’re pleased to inform you that your manuscript has been judged scientifically suitable for publication and will be formally accepted for publication once it meets all outstanding technical requirements.

Kind regards,

Job Fransen

Academic Editor

PLOS ONE

Additional Editor Comments (optional):

Congratulations on the acceptance of your work. I am following the recommendation made by the last reviewer, who has now recommended acceptance of your work. Thanks for addressing their, and the other suggestions. I think the criticisms that were made in those reviews were valid, but all have now been addressed sufficiently.

Reviewers' comments:

Reviewer's Responses to Questions

**Comments to the Author**

1. If the authors have adequately addressed your comments raised in a previous round of review and you feel that this manuscript is now acceptable for publication, you may indicate that here to bypass the “Comments to the Author” section, enter your conflict of interest statement in the “Confidential to Editor” section, and submit your "Accept" recommendation.

Reviewer #3: All comments have been addressed

2. Is the manuscript technically sound, and do the data support the conclusions?

Reviewer #3: Yes

3. Has the statistical analysis been performed appropriately and rigorously? 

Reviewer #3: Yes

4. Have the authors made all data underlying the findings in their manuscript fully available?

Reviewer #3: Yes

5. Is the manuscript presented in an intelligible fashion and written in standard English?

Reviewer #3: Yes

6. Review Comments to the Author

Reviewer #3: My comments have been addressed appropriately.

I thank the authors for their clear responses and congratulate them with this excellent study.

7. PLOS authors have the option to publish the peer review history of their article (what does this mean? ). If published, this will include your full peer review and any attached files.

**Do you want your identity to be public for this peer review?** For information about this choice, including consent withdrawal, please see our Privacy Policy .

Reviewer #3: **Yes: ** Pieter Vansteenkiste

---

## [Editor Report · Acceptance letter]

PONE-D-25-10199R2

PLOS ONE

Dear Dr. Kanatschnig,

I'm pleased to inform you that your manuscript has been deemed suitable for publication in PLOS ONE. Congratulations! Your manuscript is now being handed over to our production team.

Kind regards,

on behalf of

Dr. Job Fransen

Academic Editor

PLOS ONE